# Batch Normalization Orthogonalizes Representations in Deep Random Networks

**Hadi Daneshmand**
INRIA Paris
seyed.daneshmand@inria.fr

**Amir Joudaki**
ETH Zurich
amir.joudaki@inf.ethz.ch

**Francis Bach**
INRIA-ENS-PSL Paris
francis.bach@inria.fr

## Abstract

This paper underlines a subtle property of batch-normalization (BN): Successive batch normalizations with random linear transformations make hidden representations increasingly orthogonal across layers of a deep neural network. We establish a non-asymptotic characterization of the interplay between depth, width, and the orthogonality of deep representations. More precisely, under a mild assumption, we prove that the deviation of the representations from orthogonality rapidly decays with depth up to a term inversely proportional to the network width. This result has two main implications: 1) Theoretically, as the depth grows, the distribution of the representation –after the linear layers– contracts to a Wasserstein-2 ball around an isotropic Gaussian distribution. Furthermore, the radius of this Wasserstein ball shrinks with the width of the network. 2) Practically, the orthogonality of the representations directly influences the performance of stochastic gradient descent (SGD). When representations are initially aligned, we observe SGD wastes many iterations to orthogonalize representations before the classification. Nevertheless, we experimentally show that starting optimization from orthogonal representations is sufficient to accelerate SGD, with no need for BN.

## 1 Introduction

Batch Normalization (BN) (Ioffe and Szegedy, 2015) enhances training across a wide range of deep network architectures and experimental setups (He et al., 2016; Huang et al., 2017; Silver et al., 2017). The practical success of BN has inspired research into the underlying mechanism of BN (Santurkar et al., 2018; Karakida et al., 2019; Arora et al., 2019b; Bjorck et al., 2018). BN influences first-order optimization methods by avoiding the rank collapse in deep representation (Daneshmand et al., 2020), direction-length decoupling of optimization (Kohler et al., 2018), influencing the convergence of the steepest descent (Arora et al., 2019b; Bjorck et al., 2018), and smoothing the optimization objective function (Santurkar et al., 2018; Karakida et al., 2019). However, the benefits of BN go beyond its critical role in optimization. For example, Frankle et al. (2021) shows that BN networks with random weights also achieve surprisingly high performance after only minor adjustments of their weights. This striking result motivates us to study the representational power of random networks with BN.

We study hidden representations across layers of a laboratory random BN with linear activations. Consider a batch of samples passing through consecutive BN and linear layers with Gaussian weights. The representations of these samples are perturbed by each random linear transformation, followed by a non-linear BN. At first glance, the deep representations appear unpredictable

35th Conference on Neural Information Processing Systems (NeurIPS 2021).

after many stochastic and non-linear transformations. Yet, we show that these transformations orthogonalize the representations. To prove this statement, we introduce the notion of "orthogonality gap", defined in Section 3, to quantify the deviation of representations from a perfectly orthogonal representation. Then, we prove that the orthogonality gap decays exponentially with the network depth and stabilizes around a term inversely related to the network width. More precisely, we prove

$$\mathbb{E}\left[\text{orthogonality gap}\right] = \mathcal{O}\left((1-\alpha)^{\text{depth}} + \frac{\text{batchsize}}{\alpha\sqrt{\text{width}}}\right)$$

holds for $\alpha > 0$ that is an absolute constant under a mild assumption. In probability theoretic terms, we prove stochastic stability of the Markov chain of hidden representations (Kushner, 1967; Kushner and Yin, 2003; Khasminskii, 2011). The orthogonality of deep representations allows us to prove that the distribution of the representations after linear layers contracts to a Wasserstein-2 ball around isotropic Gaussian distribution as the network depth grows. Moreover, the radius of the ball is inversely proportional to the network width. Omitting details, we prove the following bound holds:

$$\text{Wasserstein}_2(\text{representations}, \text{Gaussian})^2 = \mathcal{O}\left((1-\alpha)^{\text{depth}}(\text{batchsize}) + \frac{(\text{batchsize})^2}{\alpha\sqrt{\text{width}}}\right).$$

The above equation shows how depth, width, and batch size, interact with the Gaussian approximation of the representations. Since the established rate is exponential with depth, the distribution of the representations stays in a Wasserstein ball around isotropic Gaussian distribution after a few layers. Thus, BN not only stabilizes the distribution of the representations, which is its main promise (Ioffe and Szegedy, 2015), but also enforces Gaussian isotropic distribution in deep layers.

There is growing interest in bridging the gap between neural networks, as the most successful parametric methods for learning, and Gaussian processes and kernel methods, as well-understood classical models for learning (Jacot et al., 2018; Matthews et al., 2018; Lee et al., 2019; Bietti and Mairal, 2019; Huang et al., 2014). This link is inspired by studying random neural networks in the asymptotic regime of infinite width. The seminal work by Neal (1996) sparks that a single-layer network resembles a Gaussian process as its width goes to infinity. However, increasing the depth may significantly shift the distribution of the representations away from Gaussian (Ioffe and Szegedy, 2015). This distributional shift breaks the link between Gaussian processes and deep neural networks. To ensure Gaussian representations, Matthews et al. (2018) suggests increasing the width of the network proportional to the network depth. Here, we show that BN ensures Gaussian representations even for deep networks with *finite* width. This result bridges the link between deep neural networks and Gaussian processes in the regime of finite width. Many studies rely on deep Gaussian representations in an infinite width setting (Yang et al., 2019; Schoenholz et al., 2017; Pennington et al., 2018; Klambauer et al., 2017; De Palma et al., 2019). Our non-asymptotic Gaussian approximation can be incorporated into their analysis to extend these results to the regime of finite width.

Since training starts from random networks, representations in these networks directly influence training. Hence, recent theoretical studies has investigated the interplay between initial hidden representations and training (Daneshmand et al., 2020; Bjorck et al., 2018; Frankle et al., 2021; Schoenholz et al., 2017; Saxe et al., 2014; Bahri et al., 2020). In particular, it is known that hidden representations in random networks *without BN* become correlated as the network grows in depth, thereby drastically slowing training (Daneshmand et al., 2020; He et al., 2016; Bjorck et al., 2018; Saxe et al., 2014). On the contrary, we prove that deep representations in networks *with BN* are almost orthogonal. We experimentally validate that initial orthogonal representations can save training time that would otherwise be needed to orthogonalize them. By proposing a novel initialization scheme, we ensure the orthogonality of hidden representations. Such an initialization effectively avoids the training slowdown with depth for vanilla networks, with no need for BN. This observation further motivates studying the inner workings of BN to replace or improve it for deep learning. Theoretically, we made the following contributions:

1. For MLPs with batch normalization, linear activation, and Gaussian weights, we prove that representations across layers become increasingly orthogonal up to a constant inversely proportional to the network width.

2. Leveraging the orthogonality, we prove that the distribution of the representations contracts to a Wasserstein ball around a Gaussian distribution as the depth grows. Up to the best of

our knowledge, this is the first *non-asymptotic* Gaussian approximation for deep neural networks with finite width.

Experimentally, we made the following contribution[1]:

3. Inspired by our theoretical understanding, we propose a novel weight initialization for standard neural networks that ensure orthogonal representations without BN. Experimentally, we show that this initialization effectively avoids training slowdown with depth in the absence of BN.

## 2 Preliminaries

### 2.1 Notations

Akin to Daneshmand et al. (2020), we focus on a Multi-Layer Perceptron (MLP) with batch normalization and linear activation. Theoretical studies of linear networks is a growing research area (Saxe et al., 2014; Daneshmand et al., 2020; Bartlett et al., 2019; Arora et al., 2019a). When weights are initialized randomly, linear and non-linear networks share similar properties such as the rank collapse issue studied in (Daneshmand et al., 2020). For ease of analysis, we assume activations are linear. Yet, we will argue that our findings extend to non-linear in Appendix G.

We use $n$ to denote batch size, and $d$ to denote the width across all layers, which we further assume is larger than the batch size $d \geq n$. Let $H_\ell \in \mathbb{R}^{d \times n}$ denote representations for $n$ samples in layer $\ell$, with $H_0 \in \mathbb{R}^{d \times n}$ corresponding to $n$ input samples in the batch with $d$ features. Successive representations are connected by Gaussian weight matrices $W_\ell \sim \mathcal{N}(0, I_d/d)$ as

$$H_{\ell+1} = \frac{1}{\sqrt{d}} BN(W_\ell H_\ell), \quad BN(M) = \text{diag}(MM^\top)^{-1/2} M, \tag{1}$$

where diag(.) zeros out off-diagonal elements of its input matrix, and the scaling factor $1/\sqrt{d}$ ensures that all matrices $\{H_k\}_k$ have unit Frobenius norm (see Appendix 2). The BN function in Eq. (1) differs slightly from the commonly used definition for BN as the mean correction is omitted. However, Daneshmand et al. (2020) shows this difference does not change the network properties qualitatively. Appendix G further discusses about the role of the mean correction term in our analysis. Readers can find all the notations in Appendix Table 2.

### 2.2 The linear independence of hidden representations

Daneshmand et al. (2020) observe that if inputs are linearly independent, then their hidden representations remain linearly independent in all layers as long as $d = \Omega(n^2)$. Under technical assumptions, Daneshmand et al. (2020) establishes a lower-bound on the average of the rank of hidden representations over infinite layers. Based on this study, we assume that the linear independence holds and build our analysis upon that. This avoids restating the technical assumptions of Daneshmand et al. (2020) and also further technical refinements of their theorems. The next assumption presents the formal statement of the linear independence property.

**Assumption $\mathcal{A}_1(\alpha, \ell)$.** *There exists an absolute positive constant $\alpha$ such that the minimum singular value of $H_k$ is greater than (or equal to) $\alpha$ for all $k = 1, \ldots, \ell$.*

The linear independence of the representations is a shared property across all layers. However, the representations constantly change when passing through random layers. In this paper, we mathematically characterize the dynamics of the representations across layers.

## 3 Orthogonality of deep representations

### 3.1 A warm-up observation

To illustrate the difference between BN and vanilla networks, we compare hidden representations of two input samples across the layers of these networks. Figure 1 plots the absolute value of

---

[1]Implementations are available at https://github.com/hadidaneshmand/batchnorm21.git

cosine similarity of these samples across layers. This plot shows a stark contrast between vanilla and BN networks: While representations become increasingly orthogonal across layers of a BN network, they become increasingly aligned in a vanilla network. More specifically, we observe that BN is able to orthognalized almost aligned representations; while, the vanilla network provides almost align representation of two orthogonal samples in deep layers. While this behaviour has been theoretically studied for vanilla networks (Daneshmand et al., 2020; Bjorck et al., 2018; Saxe et al., 2014), up to the best of our knowledge, it is not theoretically analyzed for BN networks for networks with finite width regime. In the following section, we formalize and prove this orthogonalizing property for BN networks with finite widths.

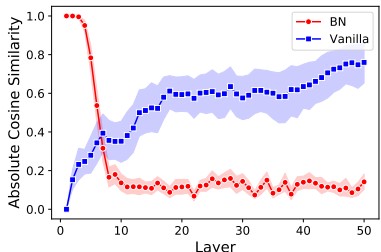

Figure 1: **Orthogonality: BN vs. vanilla networks.** The horizontal axis shows the number of layers, and the vertical axis shows the absolute value of cosine similarity between two samples across the layers ($d = 32$). Mean and 95% confidence intervals of 20 independent runs.

## 3.2 Theoretical analysis

The notion of orthogonality gap plays a central role in our analysis. Given the hidden representation $H \in \mathbb{R}^{d \times n}$, matrix $H^\top H$ constitutes inner products between representations of different samples. Note that $H^\top H \in \mathbb{R}^{n \times n}$ is different form the covariance matrix $HH^\top/n \in \mathbb{R}^{d \times d}$. The orthogonality gap of $H$ is defined as the deviations of $H^\top H$ from identity matrix, after proper scaling. More precisely, define $V : \mathbb{R}^{d \times n} \setminus \mathbf{0} \to \mathbb{R}_+$ as

$$V(H) := \left\| \left( \frac{1}{\|H\|_F^2} \right) H^\top H - \left( \frac{1}{\|I_n\|_F^2} \right) I_n \right\|_F. \tag{2}$$

The following theorem establishes a bound on the orthogonality of representation in layer $\ell$.

**Theorem 1.** *Under Assumption $\mathcal{A}_1(\alpha, \ell)$, the following holds:*

$$\mathbb{E}\left[V(H_{\ell+1})\right] \leq 2 \left( 1 - \frac{2}{3}\alpha \right)^\ell + \frac{3n}{\alpha\sqrt{d}}. \tag{3}$$

Assumption $\mathcal{A}_1$ is studied by Daneshmand et al. (2020) who note that $\mathcal{A}_1(\alpha, \infty)$ holds as long as $d = \Omega(n^2)$. If $\mathcal{A}_1$ does not hold, one can still prove that there is a function of representations that decays with depth up to a constant (see Appendix C).

The above result implies that BN is an approximation algorithm for orthogonalizing the hidden representations. If we replace $\text{diag}(M)^{-1}$ by $(M)^{-1}$ in BN formula, in Eq. (1), then all the hidden representation will become exactly orthogonal. However, computing the inverse of a *non-diagonal* $d \times d$ matrix is computationally expensive, which must repeat for all layers throughout training, and differentiation must propagate back through this inversion. The diagonal approximation in BN significantly reduces the computational complexity of the matrix inversion. Since the orthogonality gap decays at an exponential rate with depth, the approximate orthogonality is met after only a few layers. Interestingly, this yields a desirable cost-accuracy trade-off: For a larger width, the orthogonality is more accurate, due to term $1/\sqrt{d}$ in the orthogonality gap, and also the computational gain is more significant.

From a different angle, Theorem 1 proves the stochastic stability of the Markov chain of hidden representations. In expectation, stochastic processes may obey an inherent Lyapunov-type of stability

(Kushner, 1967; Kushner and Yin, 2003; Khasminskii, 2011). One can analyze the mixing and hitting times of Markov chains based on the stochastic stability of Markov chains,(Kemeny and Snell, 1976; Eberle, 2009). In our analysis, the orthogonality gap is a Lyapunov function characterizing the stability of the chain of hidden representations. This stability opens the door to more theoretical analysis of this chain, such as studying mixing and hitting times. For, example the stability can be used for mixing analysis of the chain $\{H_1, \ldots, H_\ell, \ldots\}$. Let $\pi$ denote the stationary distribution of this chain. Under a particular stochastic stability condition called geometric drift condition,

$$|\mathbb{E}\left[\varphi(H_\ell)\right] - \mathbb{E}_{H \sim \pi}\left[\varphi(H)\right]| \leq \alpha^\ell \left|\mathbb{E}\left[\varphi(H_0)\right] - \mathbb{E}_{H \sim \pi}\left[\varphi(H)\right]\right|$$

holds for $\alpha \in (0, 1)$ and measurable function $\varphi : R^{d \times n} \to R$ (see Theorem 3.6 of (Hairer, 2010)). The drift condition holds if there exists a Lyapunov function $L : R^{d \times n} \to [0, 1]$ and constant $K \geq 0$ such that the following holds:

$$E\left[L(H_{\ell+1})|H_\ell\right] \leq \gamma L(H_\ell) + K.$$

We prove that the above condition holds in Appendix Eq (12). However, such theoretical analyses may not be of interest to the machine learning community, hence we focus on the implications of these results for understanding the underpinnings of neural networks.

It is helpful to compare the orthogonality gap in BN networks to studies on vanilla networks (Bjorck et al., 2018; Daneshmand et al., 2020; Saxe et al., 2014). The function implemented by a vanilla linear network is a linear transformation as the product of random weight matrices. The spectral properties of the product of i.i.d. random matrices are the subject of extensive studies in probability theory (Bougerol et al., 2012). Invoking these results, one can check that the orthogonality gap of the hidden representations in vanilla networks rapidly increases since the rank of representations converges to a one matrix as the depth grows (Daneshmand et al., 2020). We refer readers to Appendix D for more details about vanilla networks.

Remarkably, Agarwal et al. (2021) proves if activation functions obey a self-normalization property, then a specific kernel of hidden representations becomes well-condition as the network depth grows. However, it is not clear how to impose the self-normalization property. Here, we establish the whitening for an explicitly defined normalization used in practice.

### 3.3 Experimental validations

Our experiments presented in Fig. 2a validate the exponential decay rate of $V$ with depth. In this plot, we see that $\log(V_\ell)$ linearly decreases for $\ell = 1, \ldots, 20$, then it wiggles around a small constant. Our experiments in Fig. 2b suggest that the $\mathcal{O}(1/\sqrt{d})$ dependency on width is almost tight. Since $V(H_\ell)$ rapidly converges to a ball, the average of $V(H_\ell)$ over layers estimates the radius of this ball. This plot shows how the average of $V(H_\ell)$, over 500 layers, changes with the network width, validating the $\mathcal{O}(1/\sqrt{d})$ dependency implied by Theorem 1.

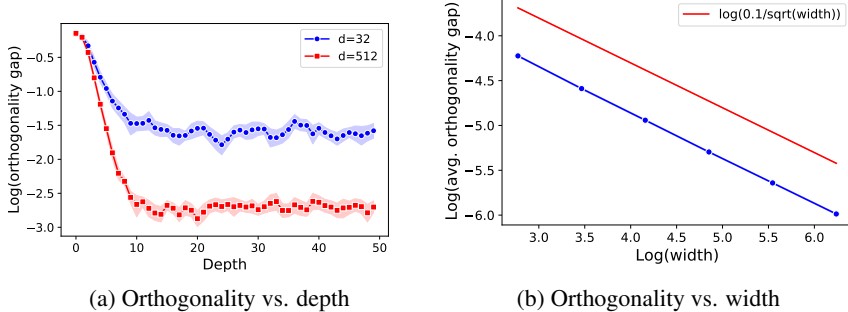

(a) Orthogonality vs. depth                 (b) Orthogonality vs. width

Figure 2: **Orthogonality gap vs. depth and width.** Left: $\log(V(H_\ell))$ vertically versus $\ell$ horizontally. Right: $\log(\frac{1}{500}\sum_{\ell=100}^{600} V(H_\ell))$ vertically versus $\log(d)$ horizontally. The chain starts from a diagonal $H_0$ with one relatively large diagonal value. This structure imposes a large orthogonality gap for $H_0$. Mean and 95% confidence interval of 20 independent runs.

Consider an input matrix $H_0 \in \mathbb{R}^{d \times n}$ containing $n$ samples in $\mathbb{R}^d$. Assuming that elements of this matrix are i.i.d. zero-mean and unit variance random variables, the minimum singular value of $H_0$

is greater than $\sqrt{d} - \sqrt{n}$. In practical applications, the batchsize used for normalization is relatively smaller than the network width, hence $\mathcal{A}_1(\Theta(1), 0)$ holds. For the intermediate representations, we also observed that $\mathcal{A}_1(\alpha, \ell)$ holds –for an $\alpha$ independent from $\ell$– as long as $d \gg n$. Let $\alpha_0$ denote the minimum singular value of $H_0$. For different values of $d$ and $n$, we check whether $\mathcal{A}_1(0.1\alpha_0, 1000)$ holds. Fig. 3 illustrates that this assumption hold for $d = \Omega(n^2)$, which is also confirmed by Daneshmand et al. (2020).

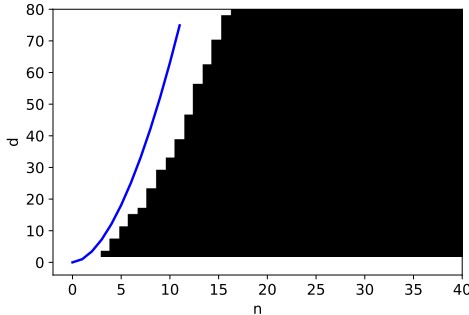

Figure 3: **Validations for** $\mathcal{A}_1$ Pixel $(n, d)$ marks whether $\mathcal{A}_1$ holds in all 10 independent runs: The black color indicates $\mathcal{A}_1(0.1\alpha_0, 1000)$ failed in at least once (where $\alpha_0$ is the minimum singular value of $H_0$). The blue curve marks $d = (n - 28)^{1.8}$ highlighting $\mathcal{A}_1$ holds for $d = \Omega(n^2)$.

## 4 Gaussian approximation

### 4.1 Orthogonality yields Gaussian approximation

The established result on the orthogonality gap allows us to show that the representations after linear layers are approximately Gaussian. If $H_\ell$ is orthogonal, the random matrix $W_\ell H_\ell$ is equal in distribution to a standard Gaussian matrix due to the invariance of standard Gaussian distribution under linear orthogonal transformations. We can formalize this notion of Gaussianity by bounding the Wasserstein-2 distance between the distribution of $W_\ell H_\ell$ and standard Gaussian distribution. Let $\mathcal{W}_2(R_1, R_2)$ denote the Wasserstein-2 distance between probability distributions of two random variables $R_1$, and $R_2$. The next lemma formally establishes the link between the orthogonality and the distribution of the representations.

**Lemma 2.** *Given $G \in \mathbb{R}^{d \times n}$ with i.i.d. zero-mean $1/d$-variance Gaussian elements, the following Gaussian approximation holds:*

$$\mathcal{W}_2\left(W_\ell H_\ell, G/\sqrt{n}\right)^2 \leq 2n\mathbb{E}\left[V(H_\ell)\right]. \tag{4}$$

Combining the above result with Theorem 1 yields the result presented in the next corollary.

**Corollary 3.** *For $G \in \mathbb{R}^{d \times n}$ with i.i.d. zero-mean $1/d$-variance Gaussian elements,*

$$\mathcal{W}_2\left(W_\ell H_\ell, G/\sqrt{n}\right)^2 \leq 4n\left(1 - \frac{2}{3}\alpha\right)^\ell + \frac{6n^2}{\alpha\sqrt{d}} \tag{5}$$

*holds under Assumption $\mathcal{A}_1(\alpha, \ell)$.*

In other words, the distribution of the representations contracts to a Wasserstein 2 ball around an isotropic Gaussian distribution as the depth grows. The radius of the Wasserstein 2 ball is at most $\mathcal{O}(1/\sqrt{\text{width}})$. As noted in the last section, $\mathcal{A}_1$ is extensively studied by Daneshmand et al. (2020) where it is shown that $\mathcal{A}_1(\alpha > 0, \infty)$ holds as long as $d = \Omega(n^2)$.

### 4.2 Deep neural networks as Gaussian processes

Leveraging BN, Corollary 3 establishes the first *non-asymptotic* Gaussian approximation for deep random neural networks. For vanilla networks, the Gaussianity is guaranteed only in the *asymptotic* regime of infinite width (Garriga-Alonso et al., 2019; Neal, 1996; Lee et al., 2019; Neal, 1996;

Hazan and Jaakkola, 2015). Particularly, Matthews et al. (2018) links vanilla networks to Gaussian processes when their width is infinite and grows in successive layers, while our Gaussian approximation holds for networks with finite width across layers. Table 1 briefly compares Gaussian approximation for vanilla and BN networks.

| Network | Width | Depth | Distribution of Outputs |
|---------|-------|-------|------------------------|
| Vanilla MLP | infinite | finite | Converges to Gaussian as width $\to \infty$ |
| Vanilla Convnet | infinite | finite | Converges to Gaussian as width $\to \infty$ |
| BN MLP (Cor. 3) | (in)finite | (in)finite | In a $\mathcal{O}(\text{width}^{-1/4})$-$\mathcal{W}_2$ ball around Gaussian |

Table 1: **Distribution of representations in random vanilla and BN networks.** For the convolutional network, the width refers to the number of channels. Results for Vanilla MLPs and Vanilla convolution networks are establish by Matthews et al. (2018), and Garriga-Alonso et al. (2019), respectively. Remarkably, Corollary 3 holds for MLP with linear activations.

The link between Gaussian processes and infinite-width neural networks has inspired several studies to rely on Gaussian representations in deep random networks (Klambauer et al., 2017; De Palma et al., 2019; Pennington et al., 2018; Schoenholz et al., 2017; Yang et al., 2019). Assuming the representations are Gaussian, Klambauer et al. (2017) designed novel activation functions that improve the optimization performance, De Palma et al. (2019) studies the sensitivity of random networks, Pennington et al. (2018) highlights the spectral universality in random networks, Schoenholz et al. (2017) studies information propagating through the network layers, and Yang et al. (2019) studies gradients propagation through the depth. Indeed, our analysis implies that including BN imposes the Gaussian representations required for these analyses. Although our result is established for linear activations, we conjecture a similar result holds for non-linear MLPs (see Appendix G).

## 5 The orthogonality and optimization

In the preceding sections, we elaborated on the theoretical properties of BN networks in controlled settings. In this section, we demonstrate the practical applications of our findings. In the first part, we focus on the relationship between depth and orthogonality. Increasing depth drastically slows the training of neural networks with BN. Furthermore, we observe that as depth grows, the training slowdown highly correlates with the orthogonality gap. This observation suggests that SGD needs to orthogonalize deep representations in order to start classification. This intuition leads us to the following question: If orthogonalization is a prerequisite for training, can we save optimization time by starting from orthogonal representations? To test this experimentally, we devised a weight initialization that guarantees orthogonality of representations. Surprisingly, even in a network without BN, our experiments showed that this initialization avoids the training slow down, affirmatively answering the question.

Throughout the experiments, we use vanilla MLP (without BN) with a width of 800 across all hidden layers, ReLU activation, and used Xavier's method for weights intialization (Glorot and Bengio, 2010). We use SGD with stepsize 0.01 and batch size 500 and for training. The learning task is classification with cross entropy loss for CIFAR10 dataset (Krizhevsky et al., 2009, MIT license). We use PyTorch (Paszke et al., 2019, BSD license) and Google Colaboratory platform with a single Tesla-P100 GPU with 16GB memory in all the experiments. The reported orthogonality gap is the average of the orthogonality gap of representation in the last layer.

### 5.1 Orthogonality correlates with optimization performance

In the first experiment, we show that the orthogonality of representations at the initialization correlates with optimization speed. For networks with 15, 30, 45, 60, and 75 widths, we register training loss after 30 epochs and compare it with the initial orthogonality gap. Figure 4a shows the training loss (blue) and the initial orthogonality gap (red) as a function of depth. We observe that representations are more entangled, i.e., orthogonal, when we increase depth, coinciding with the training slowdown. Intuitively, the slowdown is due to the additional time SGD must spend to orthogonalize the representations before classification. In the second experiment, we validate this intuitive argument by tracking the orthogonality gap during training. Figure 4b plots the orthogonality gap of

output and training loss for a network with 20 layers. We observe that SGD updates are iteratively orthogonalizing representations, marked by the reduction in the orthogonality gap.

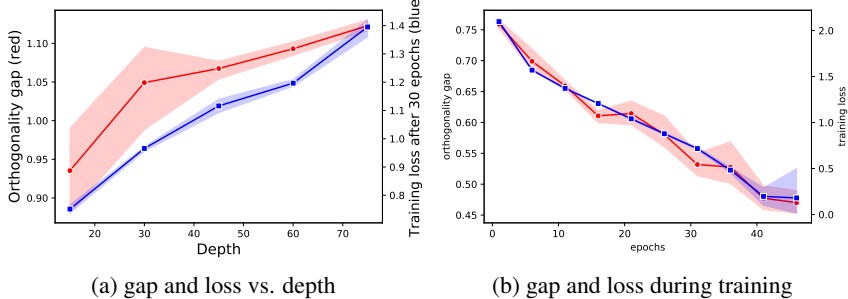

(a) gap and loss vs. depth        (b) gap and loss during training

Figure 4: **Orthogonality and Optimization** Left: the orthogonality gap at initialization (red, left axis) and the training loss after 30 epochs (blue, right axis) with depth. Right: the orthogonality gap (red, left axis) and the training loss in each epoch (blue, right axis). Mean and 95% confidence interval of 4 independent runs.

## 5.2   Learning with initial orthogonal representations

We have seen that the slowdown in SGD for deeper networks correlates with the orthogonality gap before training. Here we show that by preemptively orthogonalizing representations, we avoid the slowdown with depth. While in MPL with linear activations, hidden representations remain orthogonal simply by taking orthogonal matrices as weights (Pennington et al., 2018; Saxe et al., 2014), the same does not hold for networks with non-linear activations, such as ReLU. To enforce the orthogonality in the absence of BN, we introduce a dependency between weights of successive layers that ensures deep representations remain orthogonal. More specifically, we incorporate the SVD decomposition of the hidden representation of each layer into the initialization of the subsequent layer. To emphasis this dependency between layers and to distinguish it from purely orthogonal weight initialization, we refer to this as *iterative orthogonalization*.

We take a large batch of samples $n \geq d$, as the input batch for initialization. Let us assume that weights are initialized up to layer $W_{\ell-1}$. To initialize $W_\ell$, we compute SVD decomposition of the representations $H_\ell = U_\ell \Sigma_\ell V_\ell^\top$ where matrices $U_\ell \in \mathbb{R}^{d \times d}$ and $V_\ell \in \mathbb{R}^{n \times d}$ are orthogonal. Given this decomposition, we initialize $W_\ell$ by

$$W_\ell = \frac{1}{\|\Sigma_\ell^{1/2}\|_F} V_\ell' \Sigma_\ell^{-1/2} U_\ell^\top, \tag{6}$$

where $V_\ell' \in \mathbb{R}^{d \times d}$ is an orthogonal matrix obtained by slicing $V_\ell \in \mathbb{R}^{n \times d}$. Notably, the inverse in the above formula exists when $n$ is sufficiently larger than $d$ [2]. It is easy to check that $V(W_\ell H_\ell) < V(H_\ell)$ holds for the above initialization (see Appendix F), similar to BN. By enforcing the orthogonality, this initialization significantly alleviates the slow down of training with depth (see Fig. 5), with no need for BN. This initialization is not limited to MLPs. In Appendix H, we compare iterative orthogonalization with a BN-replacement method recently proposed by Brock et al. (2021). In Appendix I, we propose a similar SVD-based initialization for convolutional networks that effectively accelerates the training of deep convolutional networks.

## 6   Discussion

To recap, we proved the recurrence of random linear transformations and BN orthogonalizes samples. Our experiments underline practical applications of this theoretical finding: starting from orthogonal representations effectively avoids the training slowdown with depth for MLPs. Based on our experimental observations in Appendix I, we believe that this result extends to standard convolution networks used in practice. In other words, a proper initialization ensuring the orthogonality

---

[2]We may inductively assume that $H_\ell$ is almost orthogonal by the choice of $W_1, \ldots, W_{\ell-1}$. Thus, $\Sigma_\ell$ is invertible.

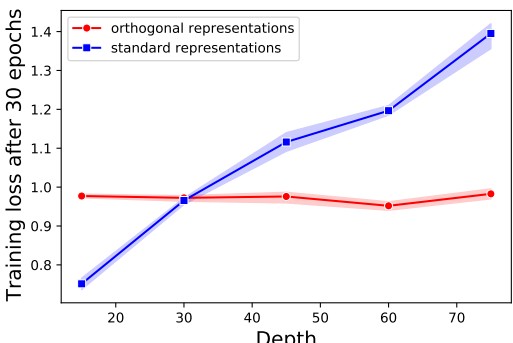

Figure 5: **Iterative orthogonalization.** Horizontal axis: depth. Vertical axis: the training loss after 30 epochs for Xavier's initialization (blue), our initialization (red). Mean and 95% confidence interval of 4 independent runs.

of hidden representations may replace BN in neural architectures. This future research direction has the potentials to boost the training of deep neural networks and change benchmarks in deep learning.

Although our theoretical bounds hold for MLPs with linear activations, our experiments confirm similar orthogonal stability for various neural architectures. Appendix Figure 6 demonstrates this stability for MLPs with ReLU activations and a convolutional networks. This plot compares the evolution of the orthogonality gap, through layers, for BN networks with vanilla networks. For vanilla networks, we observe that the gap increases with depth. On the contrary, the gap decreases by adding BN layers and stabilizes at a term that is constant with regard to depth. Based on our observations, we conjecture that hidden representations of modern BN networks obey similar stability. A more formal statement of the conjecture is presented in Appendix G.

Lubana et al. (2021) experimentally compares properties of different normalization techniques including layer normalization (LN) (Ba et al., 2016). According to this study, LN does not necessarily orthogonalize the outputs of deep neural networks. Hence, more theoretical studies are required to understand the essence of different normalization techniques in deep learning.

## Acknowledgements

We thank Gideon Dresdner, Vincent Fortuin, and Ragnar Groot Koerkamp for their helpful comments and discussions. This work was funded in part by the French government under management of Agence Nationale de la Recherche as part of the "Investissements d'avenir" program, reference ANR-19-P3IA-0001(PRAIRIE 3IA Institute). We also acknowledge support from the European Research Council (grant SEQUOIA 724063).

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
