# Appendix

## A    Preliminaries

### A.1    Notations

Let $v, w \in \mathbb{R}^k$ then $v \odot w \in \mathbb{R}^n$ with coordinates

$$[v \odot w]_i = v_i w_i \tag{7}$$

Furthermore $v^{\otimes 2} \in \mathbb{R}^{k \times k}$ with entities

$$[v^{\otimes 2}]_{ij} = v_i v_j. \tag{8}$$

In Table 2, we summarize notations introduced previously.

| Notation | Type | Definition |
|----------|------|------------|
| $\ell$ | integer | number of layers |
| $n$ | integer | batch size |
| $d$ | integer | width of the network |
| $k$ | integer | output dimension |
| $X$ | $\mathbb{R}^{d \times n}$ | input matrix |
| $H_\ell$ | $\mathbb{R}^{d \times n}$ | hidden representations at $\ell$ (obeying Eq. (4)) |
| $BN$ | $\mathbb{R}^{d \times n} \to \mathbb{R}^{d \times n}$ | batch normalization layer (defined in Eq.(4)) |
| $\text{Law}(X)$ | | the law of random matrix $X$ |
| $\sigma_i(M)$ | $\mathbb{R}^{k_1 \times k_2} \to \mathbb{R}_+$ | the $i$th largest singular value of matrix $M$ |
| $I_k$ | $\mathbb{R}^{k \times k}$ | Identity matrix of size $k$ |
| $\mathbf{1}_k$ | $\mathbb{R}^n$ | all-ones vector |

Table 2: Notations

### A.2    The Markov chain of hidden representations

Recall the chain of the hidden representations, denoted by $\{H_\ell \in \mathbb{R}^{d \times n}\}$, obeying the following recurrence:

$$H_{\ell+1} = \frac{1}{\sqrt{d}} BN(W_\ell H_\ell), \quad BN(M) = \left(\text{diag}(MM^\top)\right)^{-1/2} M, \tag{9}$$

where $W_\ell \in \mathbb{R}^{d \times n}$ are random weight matrices with i.i.d. zero-mean Gaussian elements. It is easy to check that the Frobenius norm of $H_\ell$ is one due to the row-wise normalization:

$$
\begin{aligned}
\text{Tr}\left(BN(H)BN(H)^\top\right) &= \text{Tr}\left(\text{diag}(HH^\top)^{-1/2} HH^\top \text{diag}(HH^\top)^{-1/2}\right) \\
&= \text{Tr}\left(\text{diag}(HH^\top)^{-1} HH^\top\right) \\
&= d.
\end{aligned} \tag{10}
$$

### A.3    Lyapunov function characterizing the orthogonality

We introduce the following function serving as a Lyapunov function $\widehat{V} : \mathbb{R}^{d \times n}$ characterizing the orthogonality of the hidden representations:

$$\widehat{V}(H) = \frac{1}{n} - (\sigma_n(H))^2, \tag{11}$$

where $\sigma_n(H)$ is the minimum singular value of matrix $H$. Next lemma proves $\widehat{V}(H_\ell)$ bounds the orthogonality gap.

**Lemma 4.** *For all hidden representations $H_\ell$, the following holds:*

$$V(H_\ell) \leq 2n\widehat{V}(H_\ell).$$

*Proof.* Let $\sigma_1, \ldots, \sigma_n$ be the singular values of $H_\ell$. Given these singular values, one can compute $V(H_\ell)$ as

$$(V(H_\ell))^2 = \sum_{i=1}^{n} \left( \sigma_i^2 - \frac{1}{n} \right)^2.$$

According to Eq. (10), $\sum_{i=1}^{2} \sigma_i^2 = 1$ holds. The proof is an immediate consequence of this propery.

$$V^2(H_\ell) = \sum_{i=1}^{n} \sigma_i^4 - 2 \underbrace{\left( \sum_{i=1}^{n} \sigma_i^2 \right)}_{=1} \frac{1}{n} + \frac{1}{n}$$

$$= \sum_{i=1}^{n} \sigma_i^4 - \frac{1}{n}.$$

For a freeze $\sigma_n$, the maximum of $\sum_{i=1}^{n} \sigma_i^4 - \frac{1}{n}$ subject to $\sum_{i=1}^{n} \sigma_i^2 = 1$ is met when $\sigma_1^2 = 1 - (n-1)\sigma_n^2$ and $\sigma_2 = \cdots = \sigma_n$. Using this optimal values, we get the following bound:

$$V^2(H_\ell) \leq 2(n-1)^2 \underbrace{\left( \frac{1}{n} - \sigma_n^2 \right)^2}_{=\widehat{V}^2(H_\ell(X))}.$$

Taking the square root of both sides concludes the proof. $\qquad\square$

# B   Proof of Theorem 1

The proof of Theorem 1 relies on the following Theorem that characterizes the change of $\widehat{V}$ in consecutive layers.

**Theorem 5.** *The sequence $\{H_\ell\}$ obeys*

$$\mathbb{E}\left[ \widehat{V}(H_{\ell+1})|H_\ell \right] \leq \left( 1 - \frac{2}{3}\left( \frac{1}{n} - \widehat{V}(H_\ell) \right) \right) \widehat{V}(H_\ell) + \frac{1}{\sqrt{d}}.$$

Notably, the above result does not rely on Assumption $\mathcal{A}_1$. Assuming that Assumption $\mathcal{A}_1$ holds, we complete the proof of Theorem 1. Combining the last Theorem by this assumption, we get

$$\mathbb{E}\left[ \widehat{V}(H_{\ell+1}) \right] \leq \left( 1 - \frac{2}{3}\alpha \right) \mathbb{E}\left[ \widehat{V}(H_\ell) \right] + \frac{1}{\sqrt{d}} \tag{12}$$

Induction over $\ell$ yields

$$\mathbb{E}\left[ \widehat{V}(H_{\ell+1}(X)) \right] \leq \left( 1 - \frac{2}{3}\alpha \right)^\ell \mathbb{E}\left[ \widehat{V}(H_1) \right] + \left( \sum_{k=1}^{\ell} (1 - \frac{2}{3}\alpha)^k \right) \frac{1}{\sqrt{d}}$$

$$\leq \left( 1 - \frac{2}{3}\alpha \right)^\ell \mathbb{E}\left[ \widehat{V}(H_1) \right] + \frac{3}{2\alpha\sqrt{d}}$$

An application of Lemma 4 completes the proof:

$$\mathbb{E}\left[ V(H_{\ell+1}) \right] \leq 2n\mathbb{E}\left[ \widehat{V}(H_{\ell+1}) \right]$$

$$\leq 2\left( 1 - \frac{2}{3}\alpha \right)^\ell + \frac{3n}{2\alpha\sqrt{d}}$$

## B.1   Stability analysis without $\mathcal{A}_1$

Using Theorem 5, we can prove stability of the chain $\{H_\ell\}$ without Assumption $\mathcal{A}_1$. After rearrangement of terms in Theorem 5, we get

$$\mathbb{E}\left[ \widehat{V}(H_{\ell+1})|H_\ell \right] - \widehat{V}(H_\ell) \leq -\frac{2}{3}\widehat{V}(H_\ell)\left( \frac{1}{n} - \widehat{V}(H_\ell) \right) + \frac{1}{\sqrt{d}}$$

Taking the expectation over $H_\ell$ and average over $\ell$ yields

$$\mathbb{E}\left[\frac{1}{\ell}\sum_{k=1}^{\ell}\widehat{V}(H_k)\left(\frac{1}{n}-\widehat{V}(H_k)\right)\right] \le \left(\frac{3\mathbb{E}\left[\widehat{V}(H_0)\right]}{2\ell}\right) + \frac{3}{2\sqrt{d}}$$

## C  Proof of Theorem 5

### C.1  Spectral decomposition.

Consider the SVD decomposition of $H_\ell$ as $H_\ell = U\mathrm{diag}(\sigma)V^\top$ where $U$ and $V$ are orthogonal matrices. Given this decomposition, we get

$$W_\ell H_\ell = \underbrace{W_\ell U}_{W}\mathrm{diag}(\sigma)V^\top \tag{13}$$

Since $W_\ell$ is Gaussian and $U$ is an orthogonal matrix, entities of $W$ are also i.i.d. standard normal. We will repeatedly use the above decomposition in our analyses.

### C.2  Concentration

Consider matrix $C_{\ell+1} := H_{\ell+1}^\top H_{\ell+1}$ whose eigenvalues are $\sigma_1^2, \ldots, \sigma_n^2$. The SVD decomposition of $H_\ell$ in Eq. (13) allows us to write $C_{\ell+1}$ as

$$C_{\ell+1} = \frac{1}{d}\sum_{i=1}^{d}\left(\frac{w_i \odot \sigma}{\|w_i \odot \sigma\|_2}\right)^{\otimes 2}$$

where $w_i \in \mathbb{R}^n$ is the $i$-th row of $W$, and $\sigma \in \mathbb{R}^n$ is the vector of singular values of $H_\ell$. Thus, conditioned on $\sigma$, $C_{\ell+1}$ is an empirical average of i.i.d. random vectors. This allows us to prove that this empirical average is concentrated around its expected value. The next lemma states this concentration.

**Lemma 6.** *The following concentration always holds*

$$\mathbb{E}_{W_\ell}\|C_{\ell+1} - \mathbb{E}_{W_\ell}\left[C_{\ell+1}\right]\|^2 \le 1/d$$

*where*

$$\mathbb{E}_{W_\ell}\left[C_{\ell+1}\right] = diag(p_1(\sigma), \ldots, p_n(\sigma)), \quad p_i(\sigma) := \mathbb{E}\left[\frac{\sigma_n^2 w_n^2}{\sum_{k=1}^{n}\sigma_k^2 w_k^2}\right], \quad w_i \overset{i.i.d.}{\sim} \mathcal{N}(0,1) \tag{14}$$

The concentration of $C_{\ell+1}$ allows us to prove that the Lyapunov function $\widehat{V}(H_{\ell+1})$ is concentrated around $1/n - p_n(\sigma)$.

**Lemma 7.** *The following holds*

$$\mathbb{E}_{W_\ell}\left[\left(\widehat{V}(H_{\ell+1}) - (1/n - p_n(\sigma))\right)^2\right] \le \frac{1}{d}$$

.

The last lemma allows us to predict the value of random variable $\widehat{V}(H_{\ell+1})$ by deterministic term $1/n - p_n(\sigma)$.

### C.3  Contraction.

The decay in $\widehat{V}(H_{\ell+1})$ with $\ell$ is due to term $1/n - p_n(\sigma)$ in the last lemma. This term is less than (or equal to) $V(H_\ell)$.

**Lemma 8.** *For $p_n(\sigma)$ defined in Eq. (14), the following holds:*

$$\left(\frac{1}{n} - p_n(\sigma)\right) \le \left(1 - \frac{2}{3}\left(\frac{1}{n} - \widehat{V}(H_\ell)\right)\right)\widehat{V}(H_\ell).$$

Combining the last lemma by Lemma 7 concludes the proof of Theorem 5:

$$\mathbb{E}_{W_\ell}\left[\widehat{V}(H_{\ell+1})\right] \le \left(1 - \frac{2}{3}\left(\frac{1}{n} - \widehat{V}(H_\ell)\right)\right)\widehat{V}(H_\ell) + \frac{1}{\sqrt{d}}. \tag{15}$$

To complete the proof, we prove Lemmas 6, 7, and 8.

## C.4 Proof of Lemma 6

Given the spectral decomposition of $H_\ell$ in Eq. (13), we compute element $ij$ of $C_{\ell+1}$, which is denoted by $[C_{\ell+1}]_{ij}$:

$$[C_{\ell+1}]_{ij} = [A^\top A]_{ij} = \frac{1}{d} \sum_{k=1}^{d} A_{ki} A_{kj}, \quad A_{ki} = W_{ki} \sigma_i / \sqrt{v_k}$$

where $v_k = \sum_{m=1}^{n} W_{km}^2 \sigma_m^2$. Since $W_{km}$ are zero mean and unit variace, we get

$$\mathbb{E}[C_{\ell+1}]_{ij} = 0$$

$$\mathbb{E}[C_{\ell+1}]_{ij}^2 = \frac{1}{d^2} \sum_{k=1}^{d} A_{ki}^2 A_{kj}^2 + \frac{1}{d^2} \sum_{k,k'} \underbrace{\mathbb{E}\left[A_{ki}^2 A_{kj}^2 A_{k'i}^2 A_{k'j}^2\right]}_{=0}$$

$$= \frac{1}{d^2} \sum_{k=1}^{d} A_{ki}^2 A_{kj}^2$$

holds for $i \neq j$. By summing up over $i \neq j$, we get

$$\sum_{i \neq j} \mathbb{E}[C_{\ell+1}]_{ij} = \frac{1}{d^2} \left( \sum_k \left( \sum_i A_{ki}^2 \right)^2 - \sum_{ik} A_{ki}^4 \right)$$

For the diagonal elements, we get

$$\mathbb{E}\left[C_{\ell+1}\right]_{ii} = \frac{1}{d} \sum_{k=1}^{d} \mathbb{E} A_{ki}^2$$

$$= \frac{1}{d} \sum_{k=1}^{d} \mathbb{E} A_{ki}^2$$

$$= \frac{1}{d} \sum_{k=1}^{d} \underbrace{\mathbb{E}\left[\frac{W_{ki}^2 \sigma_i^2}{\sum_{j=1}^{n} W_{kj}^2 \sigma_j^2}\right]}_{p_i(\sigma)}$$

The variance of $[C_{\ell+1}]_{ii}$ is bounded as

$$\text{var}([C_{\ell+1}]_{ii}) = \mathbb{E}\left(\frac{1}{d} \sum_{k=1}^{d} (A_{ki}^2 - p_i(\sigma))\right)^2$$

$$= \frac{1}{d^2} \sum_{k=1}^{d} (A_{ki}^2 - p_i(\sigma))^2$$

$$\leq \frac{1}{d^2} \sum_{k=1}^{d} A_{ki}^4$$

Combining results for the diagonal and off-diagonal elements yields

$$\mathbb{E}\|C_{\ell+1} - \mathbb{E}_{W_\ell}\left[C_{\ell+1}\right]\|_F^2 = \sum_{ij} \text{var}([C_{\ell+1}]_{ij})$$

$$\leq \frac{1}{d^2} \left( \sum_i A_{ki}^2 \right)^2 = 1/d$$

## C.5 Proof of Lemma 7

Notably, the eigenvalues of $C_{\ell+1}$ are squared singular values of $H_{\ell+1}$. Let $\lambda_n(C)$ denote the $n$th largest eigenvalue of matrix $C$.

$$\mathbb{E}_{W_\ell}\left[\left(\widehat{V}(C_{\ell+1}) - \left(\frac{1}{n} - p_n(\sigma)\right)\right)^2\right] \leq \mathbb{E}_{W_\ell}\left[(\lambda_n(C_{\ell+1}) - \lambda_n(\mathbb{E}_{W_\ell}[C_{\ell+1}]))^2\right]$$
$$= \mathbb{E}_{W_\ell}\left[\|C_{\ell+1} - \mathbb{E}_{W_\ell}[C_{\ell+1}]\|_F^2\right]$$
$$\leq \frac{1}{d}$$

where the last inequality relies on Lemma 6.

## C.6 Proof of Lemma 8

The proof is based an application of moment generating function that allows us to compute expectations of ratios of random variables.

**Lemma 9** (Lemma 1 in (Sawa, 1972)). *Let $X_1$ be a random variable that is positive with probability one and $X_2$ be an arbitrary random variable. Suppose that there exists a joint moment generating function of $X_1$ and $X_2$:*

$$\phi(\theta_1, \theta_2) = \mathbb{E}\left[\exp(\theta_1 X_1 + \theta_2 X_2)\right]$$

*for $\theta_1 \leq \epsilon$ and $|\theta_2| < \epsilon$ where $\epsilon$ is some positive constant. Then*

$$\mathbb{E}\left[\frac{X_2}{X_1}\right] = \int_{-\infty}^0 \left[\frac{\partial\phi(\theta_1, \theta_2)}{\partial\theta_2}\right]_{\theta_2=0} d\theta_1$$

To estimate $p_n(\sigma)$, we set $X_2 := \sigma_i^2 w_i^2$ and $X_1 = \sum_j \sigma_j^2 w_j^2$, which obtains

$$\phi(\theta_1, \theta_2) = \mathbb{E}\left[\exp(\theta_1 X_1 + \theta_2 X_2)\right]$$
$$= (2\pi)^{-n/2} \int_{-\infty}^\infty \exp((\theta_1 + \theta_2)\sigma_i^2 w_i^2 + \sum_{j\neq i}\theta_1\sigma_j^2 w_j^2)\exp(-\sum_k w_k^2/2)dw$$
$$= (2\pi)^{-n/2} \int_{-\infty}^\infty \exp((-0.5 + (\theta_1 + \theta_2)\sigma_i^2) w_i^2)dw_i \left(\prod_{j\neq i}\int_{-\infty}^\infty \exp((-0.5 + \theta_1\sigma_j^2) w_j^2)dw_j\right)$$
$$= \frac{1}{\sqrt{1 - 2(\theta_1 + \theta_2)\sigma_i^2}}\left(\prod_{j\neq i}\frac{1}{\sqrt{1 - 2\theta_1\sigma_j^2}}\right).$$

Taking derivative with respect to $\theta_2$ yields

$$\frac{\partial\phi}{\theta_2}(\theta_1, 0) = \frac{\sigma_i^2}{(1 - 2\theta_1\sigma_i^2)^{3/2}}\left(\prod_{j\neq i}\frac{1}{\sqrt{1 - 2\theta_1\sigma_j^2}}\right)$$

Using the result of the last lemma, we get

$$p_i(\sigma) = \int_{-\infty}^0 \frac{\sigma_i^2}{(1 - 2\theta\sigma_i^2)}\left(\prod_j\frac{1}{\sqrt{1 - 2\theta\sigma_j^2}}\right)d\theta$$

Therefore,

$$p_n(\sigma) = \sigma_n^2 f_n(\sigma), \quad f_n(\sigma) := \int_{-\infty}^0 \frac{d\theta}{(1 - 2\theta\sigma_n^2)^{3/2}\prod_{j\neq n}(1 - 2\theta\sigma_j^2)^{1/2}} \tag{16}$$

Since $\sum_{i=1}^{n} \sigma_i^2 = 1$ holds (see Eq. (10)), $f_n(\sigma)$ in the above formulation is minimized when the $\sigma_j^2$s are all equal for all $j \neq n$. Let $\sigma_n^2 := 1/n - \delta$ and $\sigma_j^2 := 1/n + \delta/(n-1)$ for all $j \neq i$. This allows us to establish a lowerbound on $f_n(\sigma)$ as

$$f_n(\sigma) \geq \int_0^\infty \underbrace{\left(1 + 2\theta\left(\frac{1}{n} - \delta\right)\right)^{-\frac{3}{2}} \left(1 + 2\theta\left(\frac{1}{n} + \frac{\delta}{n-1}\right)\right)^{-\frac{n-1}{2}}}_{g(\delta):=} d\theta \qquad (17)$$

Next lemma proves that $g(\delta)$ is a convex function for $\delta \in [0, 1/n]$.

**Lemma 10.** *The function $g(\delta)$, which is defined in Eq. (17), is a convex function on domain $\delta \in [0, 1/n]$.*

The convexity of $g(\delta)$ yields

$$g''(\delta) \geq 0 \; \forall \delta \implies g(\delta) \geq g(0) + \delta g'(0)$$

The above bound allow us to bound $f_n(\sigma)$ as

$$f_n(\sigma) \geq \int_0^\infty g(0)d\theta + \delta \int_0^\infty 2\theta\left(1 + \frac{2\theta}{n}\right)^{-\frac{n}{2}-2} d\theta$$

$$= 1 + \frac{2\delta n}{n+2}$$

Note that we use integration by parts to compute the above integrals. Recall $\delta = \frac{1}{n} - \sigma_n^2$. Combining the above inequality by Eq. (16) concludes the proof of the Lemma 8:

$$\frac{1}{n} - p_n(\sigma) \leq \left(1 - \frac{2n}{(n+2)}\sigma_n^2\right)\left(\frac{1}{n} - \sigma_n^2\right)$$

### C.7  Proof of Lemma 10

We can show convexity of $g(\delta)$ by showing $g''(\delta) \geq 0$ for all $\delta \in [0, 1/n]$. To this end, we define function $h_1(\delta)$ (for the compactness of notations) as

$$h_1(\delta) := \left(1 + 2\theta\left(\frac{1}{n} - \delta\right)\right)^{-5/2} \left(1 + 2\theta\left(\frac{1}{n} + \frac{\delta}{n-1}\right)\right)^{-\frac{n-1}{2}-1}.$$

Given $h_1$, the derivative of $g$ reads as

$$g'(\delta) = h_1(\delta)\left(-\frac{3}{2}(-2\theta)\left(1 + 2\theta\left(\frac{1}{n} + \frac{\delta}{n-1}\right)\right) - \frac{n-1}{2}\left(\frac{2\theta}{n-1}\right)\left(1 + 2\theta\left(\frac{1}{n} - \delta\right)\right)\right)$$

$$= \theta h_1(\delta)\left(3 + \theta\frac{6}{n} + \theta\delta\frac{6}{n-1} - 1 - \frac{2\theta}{n} + 2\theta\delta\right)$$

$$= \theta h_1(\delta)\underbrace{\left(2 + \theta\frac{4}{n} + \theta\delta\frac{2n+4}{n-1}\right)}_{h_2(\delta):=}$$

$$= \theta h_1(\delta)h_2(\delta).$$

One can readily check that $h_2'(\delta) \geq 0$. Hence, $h_1'(\delta) \geq 0$ ensures the convexity of $g(\delta)$. Consider the following auxiliary function

$$h_3(\delta) := \left(1 + 2\theta\left(\frac{1}{n} - \delta\right)\right)^{-7/2} \left(1 + 2\theta\left(\frac{1}{n} + \frac{\delta}{n-1}\right)\right)^{-\frac{n-1}{2}-2}.$$

Given $h_3$, we compute $h_1'$ as

$$h_1'(\delta) = h_3(\delta)\left(\left(-\frac{5}{2}\right)(-2\theta)\left(1 + 2\theta\left(\frac{1}{n} + \frac{\delta}{n-1}\right)\right) + \left(-\frac{n-3}{2}\right)\frac{2\theta}{n-1}\left(1 + 2\theta\left(\frac{1}{n} - \delta\right)\right)\right)$$

$$= h_3(\delta)\theta\left(5(1 + 2\theta/n + \frac{2\theta\delta}{n-1}) - \frac{n-3}{n-1}(1 + 2\theta/n - 2\theta\delta)\right)$$

$$= h_3(\delta)\theta\underbrace{\left(\frac{4n-2}{n-1} + 2\theta\frac{4n-2}{n(n-1)} + 2\theta\delta\frac{n+2}{n-1}\right)}_{h_4(\delta):=}$$

$$= \theta h_3(\delta)h_4(\delta).$$

Clearly we have $h_3(\delta), h_4(\delta) \geq 0$ for $\delta \in [0, \frac{1}{n}]$. Therefore, the proof is complete.

## D Results for vanilla networks

To compare networks with and without BN, next lemma formally establishes the contraction of orthogonality gap to a large value for networks withou BN.

**Lemma 11.** *Let $S_\ell = W_\ell \ldots W_1$. Then, there exists a positive constant $\delta$ such that the following holds*

$$\lim_{\ell \to \infty} \frac{1}{2\ell} \log\left(\left|V(S_\ell) - \sqrt{\frac{n-1}{n}}\right|\right) \leq -\delta. \tag{18}$$

In other words, the gap $V(S_\ell)$ converges to $\sqrt{(n-1)/n}$ with asymptotic rate $\exp(-\delta\ell)$ for the identity inputs. While, Thm. 1 proves the gap for BN networks converges to $n/\sqrt{d}$ with an exponential rate. For a sufficiently large $d$, $n/\sqrt{d} \ll \sqrt{(n-1)/n}$ holds.

*Proof.* Let $\sigma_1(\ell) \geq \sigma_2(\ell) \geq \cdots \geq \sigma_n(\ell)$ denote singular values of matrix $S_\ell$, then it is known that

$$\lim_{\ell \to \infty} \frac{1}{\ell} \log(\sigma_i^2(\ell)) = \frac{1}{2}\left(\log(2) + \Psi\left(\frac{d-i+1}{2}\right)\right) \tag{19}$$

holds where $\Psi$ is digamma function (Newman, 1986), which a monotonically decreasing function. Therefore,

$$\lim_{\ell \to \infty} \frac{1}{\ell}\left(\log(\sigma_2^2(\ell)) - \log(\sigma_1^2(\ell))\right) = -\delta < 0 \tag{20}$$

holds for $\delta > 0$ that can be exactly computed using function $\Psi$. The above inequality implies that $\sigma_1^2(\ell)$ increases (or decreases) faster than $\sigma_2^2(\ell)$ with an exponential rate. Using this result, we get the following limit for $j \neq 1$:

$$\lim_{\ell \to \infty} \frac{1}{\ell} \log\left(\frac{\sigma_j^2(\ell)}{\sum_i \sigma_i^2(\ell)}\right) \leq \lim_{\ell \to \infty} \frac{1}{\ell} \log\left(\frac{\sigma_2^2(\ell)}{\sigma_1^2(\ell)}\right) = -\delta \tag{21}$$

Furthermore,

$$\lim_{\ell \to \infty} \frac{1}{2\ell} \log\left|\frac{\sigma_1^2(\ell)}{\sum_i \sigma_i^2(\ell)} - 1\right| \leq \lim_{\ell \to \infty} \frac{1}{\ell} \log\left|\frac{n\sigma_2^2(\ell)}{\sigma_1^2(\ell)}\right| \leq -\delta + \lim_{\ell \to \infty} \log(n)/\ell \leq -\delta \tag{22}$$

Let $\sigma(\ell) = (\sigma_1^2(\ell), \ldots, \sigma_n^2(\ell)) \in \mathbb{R}^n$ and $1_n$ is the all one vector in $\mathbb{R}^n$ and $e_1 = (1, 0, \ldots, 0) \in \mathbb{R}^n$. Using triangular inequality, we get

$$\left|V(S_\ell) - \sqrt{(n-1)/n}\right| = \left|\left\|\frac{\sigma(\ell)}{\|\sigma(\ell)\|_1} - \frac{1}{n}1_n\right\| - \left\|e_1 - \frac{1}{n}1_n\right\|\right| \leq \left\|\frac{\sigma(\ell)}{\|\sigma(\ell)\|_1} - e_1\right\| \tag{23}$$

Therefore, we get

$$\lim_{\ell \to \infty} \frac{1}{\ell} \log \left( \left| V(S_\ell) - \sqrt{\frac{n-1}{n}} \right| \right)$$

$$\leq \lim_{\ell \to \infty} \left( \frac{1}{4\ell} \log \left( 2n \left( \frac{\sigma_2^2(\ell)}{\sum_j \sigma_j^2(\ell)} \right) \right) + \frac{1}{4\ell} \log \left( 2 \left( \frac{\sigma_1^2(\ell)}{\sum_j \sigma_j^2(\ell)} - 1 \right)^2 \right) \right) \quad (24)$$

which is bounded by $-\delta$ according to the established bounds in Eq. (21), and Eq. (22). $\square$

## E  Proof of Lemma 2

The main idea is based on a particular coupling of random matrices $W_\ell H_\ell$ and $G$. Consider the truncated SVD decomposition of $H_\ell$ as $H_\ell = U \mathrm{diag}(\sigma) V^\top$ where $U \in \mathbb{R}^{d \times n}$ and $V \in \mathbb{R}^{n \times n}$ are orthogonal matrices. Due to the orthogonality, the law of $W_\ell U$ is the same as those of $GV$. By coupling $W_\ell U = GV$, we get

$$\left( \mathcal{W}_2(W_\ell H_\ell, G/\sqrt{n}) \right)^2 = \inf_{\text{all the couplings}} \mathbb{E} \| W_\ell U \mathrm{diag}(\sigma) V^\top - GVV^\top/\sqrt{n} \|_F^2$$

$$\leq \mathbb{E} \| GV \left( \mathrm{diag}(\sigma) - I/\sqrt{n} \right) V^\top \|_F^2$$

$$= \mathbb{E} \mathrm{Tr}(GV \left( \mathrm{diag}(\sigma) - I/\sqrt{n} \right) V^\top V \left( \mathrm{diag}(\sigma) - I/\sqrt{n} \right) V^\top G^\top)$$

$$= \mathrm{Tr}(V \left( \mathrm{diag}(\sigma) - I/\sqrt{n} \right) \left( \mathrm{diag}(\sigma) - I/\sqrt{n} \right) V^\top \mathbb{E} \left[ G^\top G \right])$$

$$= \mathbb{E} \mathrm{Tr}(\left( \mathrm{diag}(\sigma) - I/\sqrt{n} \right) \left( \mathrm{diag}(\sigma) - I/\sqrt{n} \right) V^\top V)$$

$$= \mathbb{E} \| \left( \mathrm{diag}(\sigma) - I/\sqrt{n} \right) \|_F^2$$

$$= \mathbb{E} \left[ \sum_{i=1}^n (\sigma_i - 1/\sqrt{n})^2 \right]$$

$$= \mathbb{E} \left[ \sum_{i=1}^n \left( \sigma_i^2 - 1/n \right)^2 / \left( \sigma_i + 1/\sqrt{n} \right)^2 \right]$$

$$\leq n \mathbb{E} \left[ \sum_{i=1}^n \left( \sigma_i^2 - 1/n \right)^2 \right]$$

$$\leq n \mathbb{E} \left[ V^2(H_\ell) \right]$$

$$\leq 2n \mathbb{E} \left[ V(H_\ell) \right].$$

## F  Orthogonality gap for the iterative initialization

Recall the proposed initialization for weights based on SVD decomposition $H_\ell = U_\ell \Sigma_\ell V_\ell^\top$.

$$W_\ell = \frac{1}{\| \Sigma_\ell^{1/2} \|_F} V_\ell' \Sigma_\ell^{-1/2} U_\ell^\top.$$

Here, we show that

$$V(H_\ell) > V(W_\ell H_\ell) \quad (25)$$

holds as long as $V(H_\ell) \neq 0$ and $\mathcal{A}_1(\alpha, \ell)$ holds. Given singular values $H_\ell$, we get

$$V(H_\ell) = \sum_{i=1}^n \left( \sigma_i^2 - \frac{1}{n} \right)^2$$
$$= \sum_i \sigma_i^4 - 1/n, \quad (26)$$

where we used $\sum_{i=1}^n \sigma_i^2 = 1$ (see Eq. (2)). Now, we compute $V(H_\ell)$ using the singular values.

$$W_\ell H_\ell = \frac{1}{\| \Sigma_\ell^{1/2} \|_F} V_\ell' \Sigma_\ell^{1/2} V_\ell$$

Hence, the following holds

$$V(W_\ell H_\ell) = \sum_{i=1}^{n} \left( \frac{\sigma_i}{\sum_j \sigma_j} - \frac{1}{n} \right)^2$$

$$= \frac{1}{(\sum_{i=1}^{n} \sigma_i)^2} - 1/n$$

Combining with Eq. (26), we get

$$V(H_\ell) - V(W_\ell H_\ell) = \sum_{i=1}^{n} \sigma_i^4 - \frac{1}{(\sum_{i=1}^{n} \sigma_i)^2}.$$

To show that the right side of the above equation is positive, we need to prove

$$\sum_i \sigma_i^4 \left( \sum_{i=1}^{n} \sigma_i \right)^2 > 1 = \left( \sum_i \sigma_i^2 \right)^4$$

holds. Using Cauchy-Schwarz inequality, we get

$$\left( \sum_i \sigma_i^2 \right)^4 = \left( \sum_i \sigma_i^{3/2} \sigma_i^{1/2} \right)^4$$

$$\le \left( \sum_i \sigma_i \right)^2 \left( \sum_i \sigma_i^2 \sigma_i \right)^2$$

$$\le \left( \sum_i \sigma_i \right)^2 \left( \sum_i \sigma_i^4 \right),$$

where the equality in the above inequality is met only for $\sigma_i^2 = \frac{1}{n}$ (so $V(H_\ell) = 0$) under $\mathcal{A}_1$.

## G   Activation functions and the orthogonality

We have seen hidden representations become increasingly orthogonal through layers of BN MLPs with linear activations. We observed similar results for MLP with ReLU activations and also a simple convolution network (see Figure 6). Here, we elaborate more on the role activation functions in our analysis.

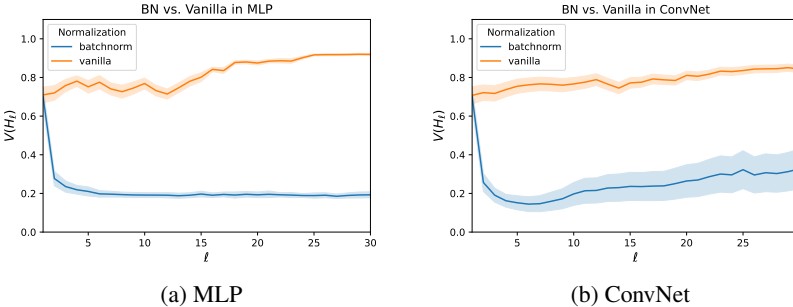

(a) MLP            (b) ConvNet

Figure 6: Orthogonality gap of layers for BatchNorm vs vanilla network with ReLU, for MLP and a basic convolutional network on CIFAR10. Batch size was set to 8, the number of hidden layers for both networks was set to 30, with 100 width for MLP and 100 channels for ConvNet in all hidden layers. For the ConvNet, the kernel size was set to 3, across all hidden layers, with zero-padding to make convolutional feature maps equal sized. The standard deviations were computed for 50 different randomly sampled batches.

For an arbitrary activation function $F$, the hidden representations make the following Markov chain:

$$Q_{\ell+1} = BN'(W_\ell F(Q_\ell)).$$

where we assume that $BN'$ makes rows of its input matrix zero-mean and unit variance (the mean correction is included).

### G.1 Conjecture

Suppose that $\{Q_\ell\}$ is *ergodict* (Eberle, 2009), and admits a unique invariant distribution denoted by $\nu$. We define function $L : \mathbb{R}^{d \times n} \to \mathbb{R}_+$ as

$$L(Q_\ell) = \left\| Q_\ell^\top Q_\ell - \mathbb{E}_{Q \sim \nu} \left[ Q^\top Q \right] \right\|_F . \tag{27}$$

We conjecture that there exists an integer $k$ such that for $\ell > k$ the following holds:

$$\mathbb{E}\left[L(Q_\ell)\right] = \mathcal{O}\left( \frac{1}{\sqrt{d}} \right) . \tag{28}$$

Figure 7 experimentally validates that the above result holds for standard activations such as ReLU, sigmoid, and tanh.

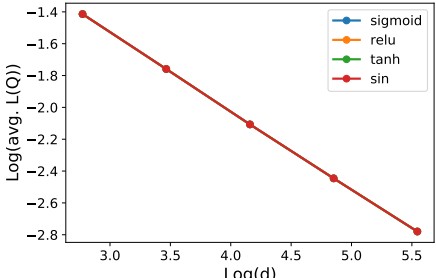

Figure 7: **Validating the conjecture.** Horizontal axis: $\log(d)$; vertical axis: $\log(\frac{1}{1000} \sum_{k=1}^{1000} L(Q_k))$. The difference between the plots is negligible, hence not visible.

### G.2 Odd activations

We observe that Theorem 1 holds for odd activations such as $\sin$ and $\tanh$ with no need to the mean correction in BN. For these activation result of Theorem 1.

$$\mathbb{E}\left[V(H_{\ell+1})\right] = \mathcal{O}\left( (1 - \alpha)^\ell + \frac{1}{\sqrt{d}} \right)$$

holds under $\mathcal{A}_1(\alpha, \ell)$. Remarkably, this is consistent with our conjecture with $V = L$ for these activations.

## H Comparisons with a BN-replacement

Here, we compare iterative orthogonalization with two baselines: (i) initialization with random orthogonal weights (Saxe et al., 2014), and (ii) adaptive gradient clipping (Brock et al., 2021).

(i). Orthogonal weights achieve orthogonal representations in deep *linear* networks. Consider a MLP whose weight are initialized by Xavier's scheme. Let the weight matrix at layer $\ell$ admit the SVD decomposition $W_\ell = U_\ell \mathrm{diag}(\sigma_\ell) V_\ell$. Then, we replace the weight matrix by the orthogonal matrix $(\mathrm{mean}(\sigma_\ell)) U_\ell V_\ell^\top$. Since ReLU networks with orthogonal weights are prone to the alignment of representations in deep layers, we observed that such an initialization does not help with the slow down of training with depth (see Fig. 8a).

(ii). Recently, Brock et al. (2021) propose an effective replacement for BN —- based on gradient clipping. Let $G_\ell$ denotes the gradient of training loss with respect to $W_\ell$. Given a clipping parameter $\lambda$, adaptive gradient clipping adjusts the norm of $G_\ell$ as

$$\widehat{G}_\ell = \begin{cases} \left( \lambda \frac{\|W_\ell\|_F^*}{\|G_\ell\|_F} \right) G_\ell & \frac{\|G_\ell\|_F}{\|W_\ell\|_F^*} \geq \lambda \\ G_\ell & \text{otherwise} \end{cases} \tag{29}$$

where $\|W_\ell\|_F^* = \max\{\|W_\ell\|_F, 10^{-2}\}$.

Fig. 8a, and 8b presents results for two different choice of clipping parameters $\lambda = 0.1$ and $\lambda = 1$, respectively. These plots demonstrate adaptive gradient clipping effectively alleviates the training slow down with depth. Yet, we observe iterative orthogonalization achieves a better training loss after 30 epochs. It is not known how the clipping enhance the training. While, iterative orthogonalization is inspired by orthogonalization of representations with BN layers, which is theoretically established in this paper.

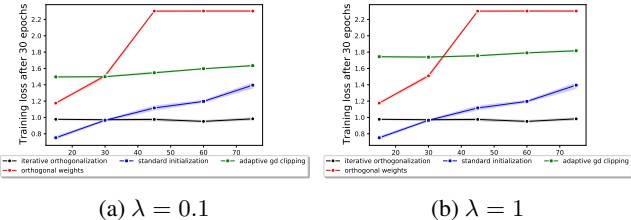

(a) $\lambda = 0.1$            (b) $\lambda = 1$

Figure 8: Iterative orthogonalization vs adaptive gradient clipping. Horizontal axis: the network depth. Vertical axis: training loss for each network after 30 SGD epochs. For more details, see Sec. 5. Mean and 95% confidence interval of 4 independent runs.

# I   Experiments for convolutional networks

The introduced initialization scheme in Section 5 extends to convolutional networks. In convolutional networks, hidden representation are in the tensor form $H_\ell \in \mathbb{R}^{d \times m \times m \times n}$ where $d$ is the number of filters and $k$ is the image dimensions. Let matrix $W_\ell^{d \times k^2}$ are weights of convolutions layer $\ell$ with kernel size $k$ and $d$ filters. A 2D convolution is a matrix multiplication combined by a projection as:

$$\text{conv2d}(W_\ell, H_\ell) = \text{fold}\left( W_\ell \underbrace{\text{unfold}(H_\ell)}_{H_\ell'} \right) \tag{30}$$

where the unfolding extract batches of $k \times k$ from images in tensor $H_\ell$, and folding operation combines the computed convolution for the extracted batches. We use the SVD decomposition $H_\ell' = U_\ell \Sigma_\ell V_\ell$ to initialize weights $W_\ell$, exactly the same as Eq. (6):

$$W_\ell = \frac{1}{\|\Sigma_\ell^{1/2}\|_F} V_\ell' \Sigma_\ell^{-1/2} U_\ell^\top,$$

where $V_\ell'$ is a slice of $V_\ell$. We experimentally validate the performance of this initialization for two convolution networks with ReLU activations consisting of 20 and 80 convolutions. Table 3 outlines the details of these neural architectures.

| The network with 20 layers | The network with 80 layers |
| :---: | :---: |
| Conv2d(3, 64)+ MaxPool2d(2) + RELU | Conv2d(3, 64)+ MaxPool2d(2) + RELU |
| Conv2d(64, 192) + MaxPool2d(2) + ReLU | Conv2d(64, 192) + MaxPool2d(2) + ReLU |
| Conv2d(192, 256) + Conv2d(256, 256) +ReLU | Conv2d(192, 256) Conv2d(256, 256) + ReLU |
| {Conv2d(256, 256)+ReLU} $\times$ 16 | {Conv2d(256, 256)+ReLU} $\times$ 76 |
| MaxPool2d(2) + ReLU | MaxPool2d(2) + ReLU |
| Dropout(0.5) + Linear(1024,4096) + ReLU | Dropout(0.5) + Linear(1024,4096) + ReLU |
| Dropout(0.5) + Linear(4096,4096) + ReLU | Dropout(0.5) + Linear(4096,4096) + ReLU |
| Linear(4096,10) | Linear(4096,10) |

Table 3: Convolutional networks used in the experiment. These neural architectures are obtained by adding layers to AlexNet (Srivastava et al., 2014). The kernel size is 3 for all the convolutional layers.

Figure 9 shows the performance of SGD (with stepsize 0.001 and batchsize 30) for the proposed initialization. This initialization slows SGD for the shallow network with 20 layers compared to

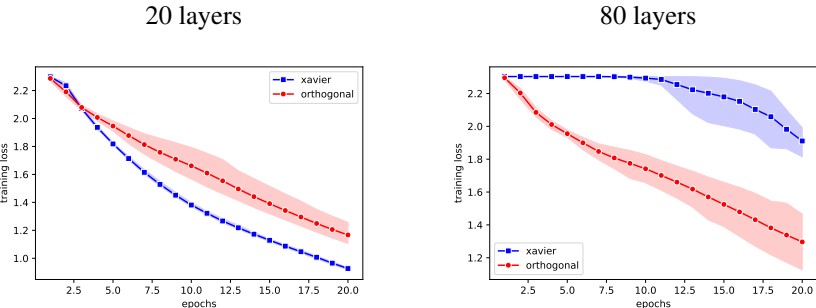

Figure 9: **Training convolutional networks.** Convergence of SGD for Xavier's initialization (Glorot and Bengio, 2010) (in blue) and also the proposed initialization method that ensures the orthogonality of hidden representations (in red). Mean and 95% confidence interval of 4 independent runs.

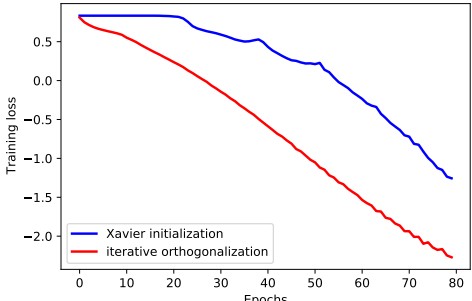

Figure 10: **Training convolutional networks.** Convergence of SGD for Xavier's initialization (Glorot and Bengio, 2010) (in blue) and also the proposed initialization method that ensures the orthogonality of hidden representations (in red). Mean of 2 independent runs.

standard Xavier's initialization (Glorot and Bengio, 2010). However, it outperforms Xavier's initialization when the depth is significantly large. This result substantiates the role of orthogonality in training. We repeated this experimented for deeper networks and more epochs. Figure 10 presents our results for a convolutional network with 90 layers and 80 epochs demonstrating orthogonal initialization boosts SGD performance even after many SGD epochs.

Although the convolution network used in this experiment is not a conventional network, this experimental result motivates further investigations of the role of orthogonality in standard convolutional networks.