# OpenReview forum: "Batch Normalization Orthogonalizes Representations in Deep Random Networks"
_NeurIPS.cc/2021/Conference — NeurIPS 2021 Spotlight_

### Official Review · Reviewer_rVTH · 2021-06-25

**Rating:** 7
**Confidence:** 3

**Summary:**

The authors study the proposition that batch normalization (BN) orthogonalizes representations in deep networks. To this end, the consider a model with linear activations and random weights. The authors prove that the "orthogonality gap", i.e. how non-orthogonal the activations are, decreases with depth when BN is used. It is also shown how the activations contract to a gaussian isotropic distribution with depth. Experimental verification is provided, and the authors also propose an initialization scheme that can be used to substitute BN.

**Main Review:**

Main Comments
---------------

Overall the paper is relatively well written. BN improving the conditioning of neural networks is folklore at this point, and it is important to provide quantitative non-asymptotic proof of this. The model the authors consider -- linear neural networks with random weights and BN without mean subtraction -- is a reasonable starting point towards a more realistic analysis. The theoretical results are reasonable, verified experimentally and interesting. The main issue with the paper is Assumption 1, which seems very strong. Consider e.g. random Guassian input. The singular values would follow the semi-circle law for random matrices, and \alpha should go to 0 as the dimensionality increases. Thus, it's not clear that it should hold for generic input. What are the reasons to think it would hold for intermediate layers? This assumption is very easy to test experimentally. Could the authors provide experimental verification that Assumption 1 holds in practice?

The contributions regarding proposing a new initialization scheme that works without BN is not very interesting in my mind. There is already plenty of work in training without normalization (see, e.g., High-Performance Large-Scale Image Recognition Without Normalization and references therein). The authors should compare against related work to show that their initialization is preferred.


Minor Comments
-----------


The authors state that "The BN function in Eq. (1) differs slightly from the commonly used definition for BN as the mean correction is omitted." This is not a minor difference, but a rather significant one. This should be highlighted more.

Fig 2b should have error bars.

It might be interesting to discuss relationships with layernorm. BN is often used for CNNs and LN is often used for MLPs which the author study here.

The authors state that they focus on "Multi-Layer Perceptron (MLP) with linear activation." This just means no activation function at all right? I find the term linear activation is confusing.

"Daneshmand et al. (2020) showes this difference does not change the network properties qualitatively." Does Daneshmand show this empirically? If so, it might only hold for the architectures they study and not in general.



Post Rebuttal Comments
-----------

The authors have clarified that the main assumption holds for the non-square case for random matrices, and also verified that it holds empirically. The assumption being strong was my main object, after the clarifications from the authors I now believe that it is reasonable. Hence, I will increase my score.



**Time Spent Reviewing:**

1

---

> ### Author Response · Authors · 2021-08-09
> **Assumption 1 is validated for random inputs and also empirically for intermediate representations**
>
>
> We thank the reviewer for their in-depth questions and feedback.
>
> > The main issue with the paper is Assumption 1, which seems very strong. Consider e.g. random Guassian input. The singular values would follow the semi-circle law for random matrices, and alpha should go to 0 as the dimensionality increases. Thus, it's not clear that it should hold for generic input.
>
> It is an elegant idea to validate Assumption 1 for random inputs. The input matrix considered in this paper is thin because the size of batch used in computing BN statistics is relatively smaller than the dimension (check the response to the first concern of reviewer XJdL for more detials on input dimension $d$ and batchsize $n$). Recall the input matrix $H_0$ is (batchsize)$\times$(dimension). Assuming elements of $H_0$ are i.i.d zero-mean and unit variance random variables, then the minimum singular value of $H_0$ is greater than $\left(\sqrt{\text{dimension}}- \sqrt{\text{batchsize}}\right)$ under mild assumptions (see *Non-asymptotic theory of random matrices: extreme singular values by
> Mark Rudelson* by Roman Vershynin), hence Assumption 1 holds true. Remarkably, Wigner's Semicircle Law holds for eigenvalues of square symmetric matrices where these eigenvalues may be negative. However, assumption 1 imposes conditions on singular values that are always non-negative.
>
> > What are the reasons to think it would hold for intermediate layers? This assumption is very easy to test experimentally. Could the authors provide experimental verification that Assumption 1 holds in practice?
>
> We agree with the reviewer that Assumption 1 can benefit from additional direct empirical validation. Let $\sigma_\ell$ denote the minimum singular value of $H_\ell$ defined in eq. (1) and $\alpha_\ell$ is the constant such that $A_1(\alpha_\ell,\ell)$ holds. The following table shows how $\sigma_\ell$ and $\alpha_\ell$ change through layers for $d=32$ and $d=256$.
>
> |$\ell$ |$\sigma_\ell$ for $d=32$ | $\alpha_\ell$ for $d=32$ | $\sigma_\ell$ for $d=256$| $\alpha_\ell$ for $d=256$ |
> |:---:|:---:|:---:|:---:|:---:|
> |0|0.002|0.002|0.008|0.008|
> |5|0.09|0.002|0.28|0.008|
> |10|0.28|0.002|0.43|0.008|
> |15|0.34|0.002|0.45|0.008|
> |50|0.34|0.002|0.45|0.008|
>
> We used $n=4$ and the reported result in the table is the average of 20 independent random linear BN networks with Gaussian weights (check recurrence of $H_\ell$ in Eq. 1). As this experiment shows, singular values initially are small since inputs are highly correlated. Yet we observe that the minimal singular value grows in depth since BN orthogonalizes the input. This empirical observation indicates that in the regime where $d\gg n$, the intermediate representations remain full rank across layers. Therefore, Assumption 1 holds for $\alpha$ equal to the minimum singular of the value of the input matrix. We will add more experimental results and intuitions about the value of $\alpha$ in Assumption 1.
>
>
> > The contributions regarding proposing a new initialization scheme that works without BN is not very interesting in my mind. There is already plenty of work in training without normalization (see, e.g., High-Performance Large-Scale Image Recognition Without Normalization and references therein). The authors should compare against related work to show that their initialization is preferred.
> Minor Comments
>
> Our experimental results bring our theoretical findings closer to piratical applications. We agree with the reviewer that there exists already interesting works on replacing BN. Yet, the proposed replacements for BN are heuristic. For example, the proposed alternative for BN in *High-Performance Large-Scale Image Recognition Without Normalization* by Brock et al. 2021 is based on adaptive gradient clipping. It is not clear how clipping replaces BN. Here, we study the inner working of BN and find that it orthogonalizes hidden representations, then we show that starting optimization from orthogonal representations accelerates optimization with no need for BN. Indeed, we propose an initialization scheme based on our theoretical findings. To highlight the importance of our theoretical findings, we will add experimental comparisons with the adaptive gradient clipping.
>
> > The authors state that "The BN function in Eq. (1) differs slightly from the commonly used definition for BN as the mean correction is omitted." This is not a minor difference, but a rather significant one. This should be highlighted more. "Daneshmand et al. (2020) showes this difference does not change the network properties qualitatively." Does Daneshmand show this empirically? If so, it might only hold for the architectures they study and not in general.
>
> We will add more details about the differences of BN with and without mean correction. For networks considered in this paper, one can show that the mean of the invariant distribution of hidden representations is zero (see Appendix C of [Daneshmand 2020]). Hence, the mean correction term for deep representations is close to zero as long as the batchsize is significantly large. We will add more details about the differences of BN with and without mean correction. Furthermore, [Daneshmand 2020] experimentally shows that that the mean correction term slightly contributes to the performance of BN (see Figure 7 of the paper) that is also confirmed by experimental results of *Weight Normalization: A Simple Reparameterization to Accelerate Training of Deep Neural Networks* by Tim Salimans et al. 2016. We will experimentally validate the significance of the mean correction in our theoretical analysis in Thm. 1.
>
> > Fig 2b should have error bars.
>
> Since this plot is in the logarithmic scale and average over many layers, the error bars are very small and not visible in the plot.
>
> > It might be interesting to discuss relationships with layernorm. BN is often used for CNNs and LN is often used for MLPs which the author study here.
>
> It is interesting to check whether layer norm also orthogonalizes the hidden representations. We will add experiments and more intuitions about this.
>
> > The authors state that they focus on "Multi-Layer Perceptron (MLP) with linear activation." This just means no activation function at all right? I find the term linear activation is confusing.
>
> We will clarify this point in the text to avoid potential confusion.

---

> > ### Comment · Reviewer_rVTH · 2021-08-10
> > **Further question**
> >
> > Thanks for the clarifications! When the dimensions batch size are the same, one would just have a square n-by-n matrix. In that setting the smallest singular value would tend to 0 as n tends to infinity right?
> >
> > Note, I have also update my score in the main review!

---

> > > ### Author Response · Authors · 2021-08-11
> > > **Response to the question**
> > >
> > > You are right: the singular value tends to 0 for large n if the matrix is square and has i.i.d. Gaussian elements with variance $1/n$. We will highlight the importance of $d \gg n$ in the paper.
> > >
> > > Thank you very much for the score update!

---

### Official Review · Reviewer_t2tK · 2021-07-15

**Rating:** 7
**Confidence:** 3

**Summary:**

This paper studies the effect of batch normalization on representations at random initialization.  They formally show that representations become increasingly orthogonal with layer depth up to a term inversely proportional to layer width. Using this, they prove that the distribution of the output contracts to a Wassertein ball around a Gaussian distribution and demonstrate empirically that orthogonal outputs save training time.  Lastly, they propose an initialization that ensures orthogonality even without batch normalization, which they empirically test.

**Limitations And Societal Impact:**

Yes.

**Main Review:**

**Originality/Significance:**
My greatest concern with this work is that it is not clear that it is a significant contribution beyond the ideas originally proposed in Daneshmand et al 2020 work "Batch Normalization Provably Avoids Rank Collapse for Randomly Initialised Deep Networks".  If we consider the contribution of this work to be in the algorithmic introduction of "iterative orthogonalization", then this work has not come close to meeting the bar for experimental benchmarking. This being said I think the ideas proposed in this paper, especially the empirical observations in section 6 (i.e. "SGD needs to orthogonalize deep representations in order to start classification") are quite interesting.  However, all the experiments provided are on very small scale networks and a much more thorough experimental section would be required to demonstrate this phenomenon holds generally.


**Clarity/Quality:**
Writing is very clear, notation is nicely introduced, and assumptions are clearly defined.  However, some experimental details are difficult to determine from both the main text and captions.



**Areas for Improvement:**

- "While the hidden representations of vanilla networks has been theoretically studied...up to the best of our knowledge, there is no theoretical analysis of BN networks".  This seems like way to strong of a claim.  For example, see Daneshmand et al 2020 work "Batch Normalization Provably Avoids Rank Collapse for Randomly Initialised Deep Networks" which you cite quite a bit or Lubana et al 2021 work "Beyond BatchNorm: Towards a General
Understanding of Normalization in Deep Learning".
- In general there seems to be quite a bit of overlap between this work and Daneshmand et al 2020.  Can you discuss the difference between your quantity of study "orthogonality gap" (equation 2 in your work) and their quantity "(stable) rank" (equation 4 in their work) and your upper bound (equation 3 in your work) and their lower bound (equation 5 in their work).
- In 4.3 Experimental validation section you are connecting your bound in equation 3 very loosely to the empirics.  Can you construct a quantitive prediction from your theory that you can validate empirically?  Of course you don't know the value of $\alpha$ but could you estimate it and then plot your bound on the same plot?  Or even plot your bound for varying levels of $\alpha$ so we can see the shape?  Making this section more quantitative would improve the empirical evidence. Also its not clear what network architecture or data sets is being used here. In general hard to determine where experimental details are.
- You write "Corollary 3 establishes the first non-asymptotic Gaussian approximation for deep random neural networks" and "the distribution of the outputs contracts to a Wassertein 2 ball around an isotropic Gaussian distribution as depth grow...the radius...is at most $\mathcal{O}(1/\sqrt{\text{width}})$".  These seem contradictory?  Only asymptotically (infinite depth/width) does the distribution of outputs according to corollary 3 become Gaussian.  Am I missing something?


**Specific Comments:**
- [line 99-100] "When weights initialized randomly, linear and non-linear networks share similar properties." => "When weights [are] initialized...share similar properties [such as...]"
- [line 105] "diag(M) zeros out off-diagonal elements" => I found this initially confusing cause in equation (1) you have $diag(MM^\intercal)$ not $diag(M)$.  Maybe just write $diag(\cdot)$ when discussing the operator?
- [line 129-130] "We used highly correlated inputs for the BN network and perfectly orthogonal inputs for the Vanilla network to contrast their differences." => I don't understand this sentence.  In the caption there is no indication of this experimental detail.  Could you elaborate here or in the caption of Fig. 1?
- [line 138] "...is different from the covariance matrix... [used in batch normalization (equation (1))]" you might want to emphasis this point?
- [equation (2)] Why write $\|I_n\|^2_F$ and not just $n$?  I would simplify this expression and put $H^\intercal H$ into numerator of first term.
- [line 161] "This stability opens the door to more theoretical analysis of this chain, such as studying mixing and hitting times" => This is still a very general statement. Can you be more specific?  What sort of questions could this stability "open the door" to answering (potentially using mixing or hitting times as an analysis tool)?
- [line 169-170] "one can readily check that the orthogonality gap of hidden representations in vanilla networks..." Could you summarize this result mathematically so we can compare to the bound in equation (3)?
- [fig. 2] "vertically versus $d$ horizontally." => dont you mean $log(d)$ horizontally?
- [fig. 4] You write earlier "to distinguish it from purely orthogonal weight initialization, we refer to this as iterative orthogonalization", but then in your label you write "orthogonal" and not "iterative orthogonalization" or "IO" or something like this.  Additionally you dont even compare to the original "orthogonal" initialization as introduced in Saxe et al.
- [Fig. 4] The yaxis label for fig4.b is "training orth"?

**Time Spent Reviewing:**

4

---

> ### Author Response · Authors · 2021-08-09
> **Major contributions are characterizing orthogonality gap and Gaussian approximation for outputs**
>
> We thank the reviewer for the helpful detailed comments. To clarify which comments are being addressed, each part is preceded by an excerpt of the reviewers' feedback.
> > My greatest concern with this work is that it is not clear that it is a significant contribution beyond the ideas originally proposed in Daneshmand et al 2020 work ...
>
> In order to address potential ambiguities between contributions of this and those of [Daneshmand 2020], here we highlight where our theoretical results are different from [Daneshmand 2020]:
> 1. **Orthogonality gap in depth:** In Thm. 1, we characterize distribution of representations in depth, while [Daneshmand 2020] state their result for average of an infinite number of layers (Thm. 2 of Daneshmand et al 2020). Due to the averaging, [Daneshmand 2020] can not explain the important interplay between the orthogonality gap and depth evident in fig.1.
>
> 2. **Orthogonality gap vs. soft rank:** While the notion of soft rank and the orthogonality gap are related, the lower bound on soft rank in Thm. 2 of [Daneshmand 2020] does not rule out highly correlated representations. More precisely, Thm. 2 of [Daneshmand 2020] obtains bound $O(1-n/d)$ on the orthogonality gap, while Thm 1. in this paper establishes bound $O(n/\sqrt{d})$ for deep representations.
> Proving orthogonality is much more challenging than preserving the rank of representations. While orthogonality of representations can be achieved perfectly by full whitening, namely by SVD decomposition, Thm. 1. proves that BN is approximately whitening input batches without the cost of exact whitening.
>
> 3. **Gaussian approximation:** Cor. 3. establishes a Gaussian approximation bound for outputs which becomes more accurate in depth, thereby linking deep neural networks to Gaussian processes. This characterization of the distribution of representations is novel and not discussed in [Daneshmand 2020].
>
>
> > ## Areas for Improvement:
> > 2)  "While the hidden ...  there is no theoretical analysis of BN networks". This seems like way to strong of a claim.
>
> Due to the averaging, [Daneshmand 2020] can not predict the dynamics of hidden representation across layers, such as the decrease of the orthogonality gap in Figure 1.
> That is why we claim that "there is no theoretical analysis of BN networks" in Section 4.1. To clarify this statement and avoid potential confusion, we will refine this statement as "there is no theoretical analysis for the observation presented in Figure 1 for the BN network".
>
> > 3) ...  Can you discuss the difference between your quantity of study "orthogonality gap" (equation 2 in your work) and their quantity "(stable) rank". ...
>
> This distinction is elaborated in **orthogonality gap vs. soft rank** in the first section.
>
> > In 4.3 Experimental validation section you are connecting your bound in equation 3 very loosely to the empiric. ...
>
> We compare the theoretical bound in Thm. 1 with the observed decrease in the orthogonality gap in Figure 2. Furthermore, we clarify that the highly correlated inputs used for the BN network allow observing the decay of the orthogonal gap across layers in Figure 2. We will add more details about generating the inputs. We observed that $\alpha$ is equal to the minimum singular value of the input matrix, and it is independent of the depth. We will present our experimental observations in this section. The network architecture in this Figure is those of our theoretical settings.
>
> > You write "Corollary 3 establishes the first non-asymptotic Gaussian approximation  ... contracts to a Wassertein 2 ball around an isotropic Gaussian distribution ...the radius...is at most $\mathcal{O}(\sqrt{1/width})$. These seem contradictory? ...
>
> To clarify, by non-asymptotic we do not imply that hidden representations are exactly Gaussian, but that the bound for Gaussianity of output in Corr.3 holds for finite width and depth. We can contrast our result with the asymptotic result of [Matthews 2018], stating that the distribution of output becomes Gaussian as width goes to infinity, without any approximation bound for a network with a finite width.
>
>
> > ## Specific Comments:
> > [line 99-100], [line 105], [line 138], [fig. 2]
>
> The typos are fixed.
>
> > [line 129-130]
>
> If inputs of the BN network are almost orthogonal, then one can not observe the decay in the orthogonality gap across the layers. Hence, we use highly correlated inputs for the BN networks. Vice versa, we use almost orthogonal inputs to show that the orthogonality gap increase in the absence of BN.
>
> > [equation (2)]
>
> We use this formulation to highlight that we use the same scaling for $H_\ell$ and $I_n$ to compute $V(H_\ell)$. We will add a note on this.
>
> > [line 161]
>
> We highlight the role of the stability in mixing analysis. Recall the Markov chain $\{ H_\ell \}$ define in Eq. 1, and let $\pi$ denote the stationary distribution of this chain. Under a particular stochastic stability condition called geometric drift condition,
>  \begin{align}
>       \left| E \left[ \varphi(H_\ell) \right] - E_{H \sim \pi} \left[ \varphi(H) \right] \right| \leq \alpha^{\ell} \left| E \left[ \varphi(H_0) \right] - E_{H \sim \pi} \left[ \varphi(H) \right] \right|
>  \end{align}
>  holds for $\alpha \in (0,1)$ and measurable function $\varphi: R^{d\times n} \to R$  (see Theorem 3.6  of *Convergence of Markov Processes* by Martin Hairer 2021). The drift condition holds if there exits a Lyapunov function $V: R^{d\times n} \to [0,1]$ and constant $K\geq 0$ such that the following holds: \begin{align}
>      E \left[ V(H_{\ell+1})| H_\ell \right] \leq \gamma V(H_{\ell}) + K.
>  \end{align}
>  We prove that the above condition holds in Eq 20 in Appendix B.  Although the drift condition is not sufficient to establish the geometric mixing bound, it is one of the required conditions. Hence, we believe our results open doors to further analysis of the chain of hidden representation. We will add the above details to the paragraph.
>
> > [line 169-170]
>
> We will add the following lemma that proves the convergences of the orthogonality gap to a large number for deep representation in vanilla networks. This result allows readers to compare the representations in BN networks with those in networks without BN.
>
>
>  *Lemma. Let $S_{\ell} = W_{1} \dots W_{\ell}$ where $W_1, \dots, W_\ell$ are random matrix whose elements are drawn i.i.d. from a Gaussian distribution. Then, there exists a positive constant $\delta$ such that the following holds* \begin{align}
>       \lim_{\ell \to \infty} \frac{1}{\ell}\log( \sqrt{V(S_\ell)}-\sqrt{(n-1)/n}) \leq -\delta.
>  \end{align}
>  In other words, the gap $\sqrt{V(S_\ell)}$ converges to $\sqrt{(n-1)/n}$ with asymptotic rate $\exp(-\gamma \ell)$. While, Thm. 1 proves the gap for BN networks converges to $n/\sqrt{d} \ll (n-1)/n$ with an exponential rate. Note the constant $\delta$ is given in the proof.
>
> *Proof of the lemma.*
> Let $\sigma_1(\ell) \geq \sigma_2(\ell) \geq  \dots \geq  \sigma_n(\ell)$ denote singular values of matrix $S_\ell$, then it is known that
>  \begin{align}
>      \lim_{\ell \to \infty} \frac{1}{\ell} \log(\sigma_i^2(\ell)) = \frac{1}{2} \left( \log(2) + \Psi\left(\frac{d-i+1}{2}\right) \right)
>  \end{align}
>  holds where $\Psi$ is digamma function (see Lyapunov exponents for products of complex Gaussian random matrices by Peter J. Forrester). Therefore,
>  \begin{align}
>       \lim_{\ell \to \infty} \frac{1}{\ell} \left( \log(\sigma_2^2(\ell)) - \log(\sigma^2_1(\ell)) \right)  = - \delta <0
>  \end{align}
>  holds for $\delta>0$ that can be exactly computed using function $\Psi$. The above inequality implies that $\sigma_1^2(\ell)$ increases (or decreases) faster than $\sigma_2^2(\ell)$ with an exponential rate.
>  Using this result, the following holds for $j\neq 1$:
>  \begin{align}
>     \lim_{\ell \to \infty} \frac{1}{\ell} \log\left(\frac{\sigma_j^2(\ell)}{\sum_{i} \sigma_i^2(\ell)}\right) \leq \lim_{\ell \to \infty} \frac{1}{\ell} \log\left(\frac{\sigma_2^2(\ell)}{ \sigma_1^2(\ell)}\right) = -\delta
>  \end{align}
>  Furthermore,
>  \begin{align}
>      \lim_{\ell \to \infty} \frac{1}{\ell} \log\left|\frac{\sigma_1^2(\ell)}{\sum_{i} \sigma_i^2(\ell)} - 1\right|
>       \leq \lim_{\ell \to \infty} \frac{1}{\ell} \log\left|\frac{n \sigma_2^2(\ell) }{ \sigma_1^2(\ell)}\right| \leq -\delta + \lim_{\ell \to \infty} \log(n)/\ell \leq -\delta
>  \end{align}
>  Let $\sigma(\ell) = (\sigma_1^2(\ell), \dots, \sigma_n^2(\ell)) \in R^n$ and $1_n$ is the all one vector in $R^n$ and $e_1 = (1,0, \dots, 0)\in R^n$.
>  Using triangular inequality, we get
>  \begin{align}
>      \left| \sqrt{V(S_\ell)} -  \sqrt{(n-1)/n} \right| & = \left|  \| \frac{\sigma(\ell)}{\| \sigma(\ell)  \|_1 }- \frac{1}{n} 1_n \| - \| e_1 -  \frac{1}{n} 1_n  \|    \right|  \leq \| \frac{\sigma(\ell)}{\| \sigma(\ell)  \|_1} - e_1 \|
>  \end{align}
>  Therefore, we get
>
>   \begin{align}
>      \lim_{\ell \to \infty} \frac{1}{\ell} \log \left( \sqrt{V(S_\ell)} - \sqrt{(n-1)/n} \right) \leq
>      \lim_{\ell \to \infty} \frac{1}{4\ell} \log \left( 2n \left(\frac{\sigma_2^2(\ell)}{\sum_{j} \sigma_j^2(\ell)} \right)^2  \right) + \frac{1}{4\ell} \log\left(2  \left( \frac{\sigma_1^2(\ell)}{\sum_{j} \sigma_j^2(\ell)} - 1 \right)^2 \right)
>  \end{align}
>  which is bounded by $-\delta$ according to the established bounds above.
>
>  > [fig. 4]
>
>   We will fix the typos. The following Table presents the requested comparison.
>
> |epoch | training loss for the proposed initialization |training loss for Saxe's initialization |
> |:---:|:---:|:---:|
> |1|2.00|2.30|
> |5|1.57|2.15|
> |10|1.42|2.12|
> |15|1.30|2.09|
> |20|1.20|2.05|
>
> We used a ReLU MLP with width 800 and 120 layers that classifies CIFAR10 using cross-entropy loss (the same setting as Figure 4). For Saxe's initialization, we use a random orthogonal matrix obtained by singular vectors of the Gaussian random matrix. The numbers in the table are the average of 4 independent runs. We will add this comparison with more details to Section 6.2.

---

> > ### Comment · Reviewer_t2tK · 2021-08-15
> > **Thank you for clarifications**
> >
> > Thank you for the detailed response and clarifications, especially with regards to the related work. After reading your response, the other reviews, and the group reviews I have updated my perspective and rating.

---

### Official Review · Reviewer_Agdx · 2021-07-18

**Rating:** 7
**Confidence:** 4

**Summary:**

This papers focuses on the analysis of batch normalization which has been attracting a lot of attention recently.  The key contribution of the paper is in the finding that batch normalization (under some rather mild and sensible assumptions) makes training samples increasingly orthogonal.

**Limitations And Societal Impact:**

All fine.

**Main Review:**

This paper is another contribution in the now rather hot area which attempts to understand better batch normalization, and NN normalization in general.  As such, it addresses a relevant and widely interesting topic.  The key findings are also rather interesting, novel, insightful, and practically important.  The bottom line demonstrated by the authors is rather simple to understand: predicated on a few mild assumptions, batch normalization makes training samples increasingly orthogonal.  This simple to understanding finding is rather insightful though and facilitates not only a better understanding of the normalization for its own sake, but also a series of practically important improvements, e.g. as the authors demonstrate, in the context of stochastic gradient descent application.  Overall, I consider the novelty and general technical quality of this paper to be well within the expected range for NIPS.  The quality of writing is also good - I have no objections to make.

Post Rebuttal:
Having considered my colleagues' reviews and the authors' response to these, I find myself no less positive about this paper and am in fact even more confident of my original positive recommendation.  Hence I see no need to alter my review or the recommendation.

**Time Spent Reviewing:**

1

---

> ### Author Response · Authors · 2021-08-09
> **We are grateful for this reviewer’s comments, and believe that their summary captures our main contributions very well.**
>
> as there are no objections by this reviewer, there are no further explanations.

---

### Official Review · Reviewer_XJdL · 2021-07-23

**Rating:** 7
**Confidence:** 3

**Summary:**

This paper considers deep linear networks with batch normalization. The main contributions are:
1. Under a linear independence condition, it is proved that the orthogonality gap is the sum of a term that decreases exponentially fast with the depth, and a term that is basically number_of_samples / sqrt{width}.
2. Under similar conditions, it is proved that the distribution of the hidden representation is close to the Gaussian distribution, and a Wasserstein distance bound is proved.
3. It is shown empirically that a larger orthogonality gap at initialization makes training slower, and that SGD decreases the orthogonality gap during training. This paper further proposes an "iterative orthogonalization" algorithm, which can enforce orthogonality and accelerate training when the depth is large.

**Limitations And Societal Impact:**

The limitations are discussed above.

**Main Review:**

I think the contributions of this paper are interesting. Intuitively, orthogonality would help with training, and it is nice to see that batch normalization can increase orthogonality. It is also interesting to see that SGD can increase orthogonality during training (even though there is no proof). The iterative orthogonalization algorithm may also be useful in practice.

I have the following concerns:
1. The main results require the width to be at least n^2, and if I understand it correctly, the width is also equal to the input dimension. It is also mentioned that the prior work by Daneshmand et al. (2020) requires all inputs to be linearly independent, which also seems to be required in this paper. These assumptions are strong.
2. As we can see in Figure 4 and 7, iterative orthogonalization always decreases the orthogonality gap, but it actually makes training slower for shallow (depth <= 20) networks.
3. In Figure 7(b), only the loss curves after 20 epochs are plotted, at which time the losses are still high. Can you also provide the loss curves after maybe 200 epochs?

Finally, the prior work "A deep conditioning treatment of neural networks" by Naman Agarwal, Pranjal Awasthi, and Satyen Kale may be related, where they studied the orthogonalization property of nonlinear activations.

**Time Spent Reviewing:**

8

---

> ### Author Response · Authors · 2021-08-09
> **Linear independence is a necessary, but a weak condition**
>
> We are thankful for the helpful comments of the reviewer.
> We will provide explanations preceded by the corresponding points raised by the reviewer and will make necessary changes to the text.
>
>
> > The main results require the width to be at least n^2, and if I understand it correctly, the width is also equal to the input dimension.
>
> It is important to note that $n$ is the *batch size* used in BN and *not the training set*. While small batch sizes like $n=16$ or $n=32$ suffice for most applications, the input dimension is often an order of magnitude larger, namely $d=32\times32=1024$ for CIFAR dataset. Therefore the assumption $d\geq n^2$ holds in many practical settings.
>
>
> > It is also mentioned that the prior work by Daneshmand et al. (2020) requires all inputs to be linearly independent, which also seems to be required in this paper. These assumptions are strong.
>
> To put this assumption into context, in the parameter regime where the input dimension is larger than batch size $d\ge n$, Assumption 1 implies that samples within each batch used for BN are linearly independent not all training samples. Note that linear independent samples may be highly correlated.
> This fact is evident in Figure 1, demonstrating that even for nearly aligned input samples, BN orthogonalizes representations across layers.
> It is worth noting that, this assumption is a necessary condition for orthogonalization of hidden representations since the recurrence of Eq. (1) can not increase the rank of representations. We also validate this assumption for random gaussian inputs in our response to the first concern of reviewer rVTH.
>
> > As we can see in Figures 4 and 7, iterative orthogonalization always decreases the orthogonality gap, but it makes training slower for shallow (depth <= 20) networks.
>
> We agree with the reviewer that the iterative orthogonalization scheme may slow training for the shallow network. But this method is designed for deep networks. This can be explained by the fact that for shallow networks, the performance of SGD is more tied to the orthogonality of inputs.
>
> > In Figure 7(b), only the loss curves after 20 epochs are plotted, at which time the losses are still high. Can you also provide the loss curves after maybe 200 epochs?
>
> A new plot similar to fig.7 but with additional epochs will be added to the appendix. The following table presents our result for 80 epochs.
>
> |epoch |  training loss for Xavier’s initialization | training loss for iterative orthogonalization. |
> |:---:|:---:|:---:|
> |1|2.30|2.28|
> |20|1.81|0.99|
> |40|1.04|0.33|
> |60|0.36|0.10|
> |80|0.10|0.06|
>
>
>
>
> > Finally, the prior work *A deep conditioning treatment of neural networks* by Naman Agarwal, Pranjal Awasthi, and Satyen Kale may be related, where they studied the orthogonalization property of nonlinear activations.
>
> We thank the reviewer for pointing out this relevant work. We will add the paper with more details to our literature review.

---

> > ### Comment · Reviewer_XJdL · 2021-08-19
> > **Thank you for your response**
> >
> > Thank you for your detailed response! I agree the assumptions are mild if the batch size is small. I will keep my score.

---

### Author Response · Authors · 2021-08-10
**Summary of contributions**

To start, we would like to thank all reviewers for their careful observations and insightful comments.

This excerpt from reviewer `rVTH` feedback captures the motivation behind this work very well
> BN improving the conditioning of neural networks is folklore at this point, and it is important to provide quantitative non-asymptotic proof of this. The model the authors consider -- linear neural networks with random weights and BN without mean subtraction -- is a reasonable starting point towards a more realistic analysis. The theoretical results are reasonable, verified experimentally and interesting.

In fact, since associating BN with improved conditioning falls short of explaining BN, finding a more precise explanation for the role of BN was the main motivation behind this study. While our theoretical setup allows us to make our observations rigorous, the experiments demonstrate that these phenomena extend well beyond artificial settings. This viewpoint is shared by reviewer `Agdx`:
> This paper ... addresses a relevant and widely interesting topic. The key findings are also rather interesting, novel, insightful, and practically important. The bottom line demonstrated by the authors is rather simple to understand: predicated on a few mild assumptions, batch normalization makes training samples increasingly orthogonal. This simple to understanding finding is rather insightful though and facilitates not only a better understanding of the normalization for its own sake, but also a series of practically important improvements, ...

From a more empirical perspective, reviewer `XJdl` shares their view on the link between orthogonality and optimization
> Intuitively, orthogonality would help with training, and it is nice to see that batch normalization can increase orthogonality. It is also interesting to see that SGD can increase orthogonality during training (even though there is no proof). The iterative orthogonalization algorithm may also be useful in practice. ...

The connection between orthogonality and optimization and a full benchmarking of *iterative orthogonalization* are exciting directions and demand follow-up work. This view is shared by reviewer `t2tk:
> ... This being said I think the ideas proposed in this paper, especially the empirical observations in section 6 (i.e. "SGD needs to orthogonalize deep representations in order to start classification") are quite interesting. However, all the experiments provided are on very small scale networks and a much more thorough experimental section would be required to demonstrate this phenomenon holds generally. ...

Finally, we would like to thank the reviewers for their constructive criticism. The points raised are addressed individually.

---

### Decision · Program_Chairs · 2021-09-27

**Decision:**

Accept (Spotlight)

**Comment:**

This paper provides a setting and analysis where batch normalization increasingly orthogonalizes data as it is mapped through a network.  Reviewers are uniformly supportive and I recommend acceptance.  Even so, a few of the reviews (and responses) were quite detailed, and I request the authors adjust the paper to clarify these points in their revisions.